# Continuous-Time Value Iteration for Multi-Agent Reinforcement Learning

**Xuefeng Wang**[1]*, **Lei Zhang**[1]*, **Henglin Pu**[1], **Ahmed H. Qureshi**[1]†, **Husheng Li**[1]†

[1]Purdue University

## Abstract

Existing reinforcement learning (RL) methods struggle with complex dynamical systems that demand interactions at high frequencies or irregular time intervals. Continuous-time RL (CTRL) has emerged as a promising alternative by replacing discrete-time Bellman recursion with differentiable value functions defined as viscosity solutions of the Hamilton–Jacobi–Bellman (HJB) equation. While CTRL has shown promise, its applications have been largely limited to the single-agent domain. This limitation stems from two key challenges: (i) conventional methods for solving HJB equations suffer from the curse of dimensionality (CoD), making them intractable in high-dimensional systems; and (ii) even with learning-based approaches to alleviate the CoD, accurately approximating centralized value functions in multi-agent settings remains difficult, which in turn destabilizes policy training. In this paper, we propose a CT-MARL framework that uses physics-informed neural networks (PINNs) to approximate HJB-based value functions at scale. To ensure the value is consistent with its differential structure, we align value learning with value-gradient learning by introducing a Value Gradient Iteration (VGI) module that iteratively refines value gradients along trajectories. This improves gradient accuracy, in turn yielding more precise value approximations and stronger policy learning. We evaluate our method using continuous-time variants of standard benchmarks, including multi-agent particle environment (MPE) and multi-agent MuJoCo. Our results demonstrate that our approach consistently outperforms existing continuous-time RL baselines and scales to complex cooperative multi-agent dynamics. Code is available at this link.

## 1 Introduction

RL has achieved remarkable success in a range of discrete-time single- and multi-agent interaction tasks, including robotic manipulation (Brunke et al., 2022), strategy games (Vinyals et al., 2019), wireless communications (Feriani & Hossain, 2021; Wang et al., 2023), and traffic coordination (Haydari & Yılmaz, 2020). However, many real-world domains are inherently continuous-time and operate at high or irregular decision frequencies, such as continuous-time state estimation in robotics (Talbot et al., 2025), autonomous driving (Brechtel et al., 2014), and market trading (Shavandi & Khedmati, 2022). Despite this, most existing RL methods formulate decision-making in *discrete-time*, where Bellman updates are computed at a fixed time interval. Such discrete-time RL (DTRL) approximates a continuous-time process by imposing a fixed discretization step, which introduces two inherent limitations. First, when the timestep is coarse, the resulting controller becomes non-smooth and leads to suboptimal or unstable behavior (Doya, 2000). Second, when the timestep is fine, the number of states and iteration steps become large, which enlarges not only memory storage but also many learning trials (Doya, 2000). In addition, as the time step $\Delta t$ approaches to 0, the Bellman operator can become ill-conditioned: the temporal-difference objective functions may be dominated by approximation noise, leading the Bellman updates to become unstable and have a large variance (Wang et al., 2020; Shilova et al., 2024). These limitations indicate that performance may critically depend on the chosen discretization method, thereby motivating continuous-time RL (CTRL) methods that avoid timestep-discretization and directly learn value functions in continuous time (Doya, 2000; Rubanova et al., 2019).

---

*Equal contribution.

†Corresponding author.

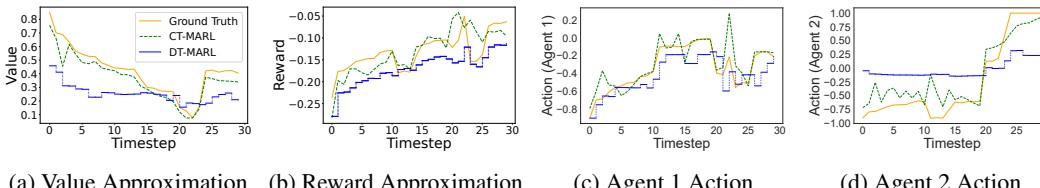

| (a) Value Approximation | (b) Reward Approximation | (c) Agent 1 Action | (d) Agent 2 Action |

Figure 1: The performance of our CT-MARL and DT-MARL is compared on a continuous-time, two-agent coupled oscillator task. In the discrete-time setting, DT-MARL trained with MADDPG can achieve near-optimal performance. However, when transferred to the continuous-time domain, MADDPG suffers from significant bias and error, resulting in poor approximations. In contrast, CT-MARL yields smoother actions, higher rewards, and more accurate value approximations, closely aligning with the analytical LQR ground truth.

Unlike Bellman operator-based DTRL (Bellman, 1966), CTRL leverages HJB Partial Differential Equation (PDEs) to compute differential value functions (Mukherjee & Liu; Shilova et al., 2024). However, solving HJB PDEs through conventional approaches (e.g., dynamic programming or level set method (Osher et al., 2004)) suffers from CoD in high-dimensional dynamical systems (Bellman et al., 1965), especially where the computational complexity grows exponentially with the state dimension in multi-agent systems. PINNs have emerged as a powerful tool to circumvent CoD (Weinan et al., 2021), and offer convergence guarantees for problems with smooth solutions (Shin et al., 2020; Ito et al., 2021). To approximate the solutions of HJB PDEs, PINNs translate the underlying physics law (e.g., PDEs) along with boundary conditions into the loss functions to refine networks.

Yet existing CTRL methods focus almost exclusively on single-agent settings and do not extend naturally to MARL. In multi-agent domains, each agent must interact with the environment while simultaneously adapting to other learning agents, introducing severe non-stationarity, making value learning substantially harder (Yu et al., 2022). Even with PINNs, accurately learning value functions under the centralized training decentralized execution (CTDE) framework remains challenging: PDE-based residual losses alone often produce biased or inaccurate value gradients, where coupled dynamics amplify approximation errors (Zhang et al., 2024b). These inaccuracies propagate through the PDE residuals, degrade value learning, and ultimately deteriorate policy optimization. To address this limitation, we propose a novel learning approach that combines PINN and VGI to optimize the value learning. The PINN component ensures the value approximations satisfy the HJB PDEs, while VGI iteratively propagates and refines the value gradient approximations along the sampled trajectories. To better understand the limitations of DTRL and advantages of our CTRL algorithms for multi-agent scenarios, we present a didactic case as shown in Fig. 1. In this simple continuous-time control task, DT-MARL fails to accurately approximate the true value functions, leading to incorrect control actions, particularly for agent 2. In contrast, our CT-MARL algorithm closely follows the ground-truth trajectory, maintains high returns, and generates accurate control actions for both agents.

Our work makes the following contributions. **(1)** We leverage PINNs to approximate differential value functions and apply them to solve *cooperative continuous-time multi-agent reinforcement learning* problems within deterministic systems, which have rarely been explored by previous studies. **(2)** We introduce a novel *value-gradient iteration* term that dynamically refines the approximations of value gradients during training. This setting improves the computational accuracy of the value gradients, accelerates learning convergence, and leads to highly accurate value approximations, enabling efficient policy learning. **(3)** We create continuous-time versions of two standard MARL benchmarks, the continuous-time MPE and continuous-time multi-agent MuJoCo. The results demonstrate that our method consistently surpasses other current CTRL baselines, highlighting the advantages of precise value-gradient learning in high-dimensional multi-agent systems.

## 2 Related Work

### 2.1 Continuous-Time Reinforcement Learning

CTRL has received increasing attention in recent years, however, most existing works focus on the single-agent settings. These studies aim to optimize policies in continuous-time domains, avoiding time discretization and yielding more accurate control actions in robotics and navigation. For instance,

Bian & Jiang (2021) proposes a continuous-time value iteration algorithm for solving HJB equations without the need for a stabilizing initial policy, while Faradonbeh & Faradonbeh (2023) introduces an HJB-based actor-critic network with theoretical guarantees for the infinite-horizon case. Similarly, Lee & Sutton (2021) develops two policy iteration algorithms that compute comprehensive solutions to HJB equations. In contrast, Jia & Zhou (2022b) introduces a temporal–difference learning based algorithm to deal with continuous-time problems via discretization. Building on this line, Jia & Zhou (2022a; 2023) uses rigorous algorithms and strong theoretical foundations to study single-agent CTRL with stochastic dynamics. Other approaches include Yang et al. (2021), which develops a robust actor–critic framework for nonlinear systems with unmodeled dynamics; Shilova et al. (2024), which leverages PINNs with $\epsilon$-scheduling to approximate value functions and empirically outperforms discrete-time RL baselines; and Yildiz et al. (2021), which integrates neural ODEs with Bayesian inference to model uncertainty in state evolution and proposes a continuous-time actor–critic algorithm that mitigates challenges such as Q-function vanishing and poor discretization. In contrast, CT-MARL remains relatively underexplored compared to the substantial studies for single-agent settings. For example, Luviano & Yu (2017) solves the multi-agent pathfinding problem using fuzzy Q-learning, while Jiang et al. (2023) proposes a model-based value iteration algorithm tailored for continuous-time multi-agent systems. Beyond these examples, only a limited number of studies have addressed CT-MARL, highlighting the importance of our contributions.

## 2.2 SOLVING HJB EQUATIONS VIA PINNS

In single-agent optimal control or multi-agent cooperative settings, value functions are characterized as the viscosity solutions to HJB equations (Crandall & Lions, 1983), which are the first-order nonlinear parabolic PDEs. However, solving HJB equations with conventional numerical methods is computationally intractable in high-dimensional settings due to CoD (Osher et al., 2004; Osher & Shu, 1991). Recent studies show that PINNs can mitigate CoD by leveraging their Monte Carlo nature when PDE solutions are smooth (Weinan et al., 2021). PINNs approximate value functions using trainable neural networks by minimizing PDE-driven loss functions, including boundary residuals (Han & Long, 2020; Han et al., 2018), PDE residuals (Bansal & Tomlin, 2021; Zhang et al., 2024a;b), and supervised data derived from numerical solvers (Nakamura-Zimmerer et al., 2021). Notably, recent studies demonstrate that integrating HJB-based PINNs with Proximal Policy Optimization (PPO) leads to improved performance over standard PPO in continuous-time single-agent MuJoCo environments (Mukherjee & Liu). However, solving CT-MARL problems through the integration of PINNs and RL remains an open and unexplored area of research.

## 3 METHODOLOGY

In this section, we present our Value Iteration via PINN (VIP) framework for solving CT-MARL problems. **1) We first formulate the problem setting** as the cooperative multi-agent deterministic system in continuous-time. By adopting a CTDE paradigm, **2) we approximate the HJB-based value function** using a PINN equipped with residual and anchor losses. To further enhance the stability and accuracy of both the value and its gradient, we incorporate additional VGI loss that iteratively refines both value and value-gradient accuracy along sampled trajectories. Building upon this critic, **3) we derive a continuous-time advantage function** directly from the HJB residual to guide decentralized policy updates, which enables each agent to update its actor in a manner consistent with continuous-time optimality conditions.

## 3.1 PROBLEM FORMULATION

In this paper, we focus on multi-agent cooperative settings. Following the continuous-time control system framework (Yildiz et al., 2021; Lee & Sutton, 2021), we formulate the continuous-time multi-agent problem as a tuple

$$\mathcal{M} = \left\langle \mathcal{X}, \{\mathcal{U}_i\}_{i=1}^{N}, N, f, r, \{t_k\}_{k \geq 0}, \rho \right\rangle. \tag{1}$$

where $\mathcal{X} \subseteq \mathbb{R}^n$ is the state space and $\mathcal{U} = \mathcal{U}_1 \times \cdots \times \mathcal{U}_N \subseteq \mathbb{R}^m$ represents the joint action space of $N$ agents. The global state and control input are represented by $x \in \mathcal{X}$ and $u \in \mathcal{U}$. Agent interactions occur over an infinite time horizon. The multi-agent system evolves according to time-invariant nonlinear dynamics defined by $\dot{x} = f(x, u)$, where $f : \mathcal{X} \times \mathcal{U} \to \mathcal{X}$ is the global dynamics function.

We define $\pi : \mathcal{X} \to \mathcal{U}$ as the decentralized joint policy $\pi = (\pi_1, \ldots, \pi_N)$. All agents share a global reward $r : \mathcal{X} \times \mathcal{U} \to \mathbb{R}$. $\rho \in (0, 1]$ is the discount factor or time-horizon scaling parameter. Unlike standard formulations that assume a fixed time step, we consider a strictly increasing sequence of decision times $\{t_k\}_{k \geq 0}$ with variable gaps $\tau_k = t_{k+1} - t_k > 0$. In this paper, we assume that $\mathcal{U}_i$ is compact and convex; $f$ is Lipschitz continuous; $r$ is Lipschitz continuous and bounded.

**Definition 1** (Value Function of Multi-agent Systems). Given $u = (u_1, \ldots, u_N)$ as a joint control input, the optimal global value function is defined as

$$V(x) = \max_{u \in \mathcal{U}} \int_t^\infty e^{-\rho(\tau - t)} r(x(\tau), u(\tau)) d\tau \tag{2}$$

## 3.2 HJB and Policy Learning

For CT-MARL problems, we build on HJB PDEs rather than discrete-time Bellman equations (Bellman, 1966), which are ill-suited to continuous-time settings. In this subsection, we explain how the HJB equations are leveraged to solve the CT-MARL problems. Specifically, we define a value network $V_\theta$ parametrized by a set of weights and biases $\theta$, and describe how the global value function $V_\theta$ is trained and how each agent's policy $\pi_{\phi_i}$ is updated, such that the overall procedure serves as a continuous-time analogue of actor–critic policy iteration. The convergence of HJB-based PINNs for learning $V_\theta$ has been established in Meng et al. (2024), providing theoretical support for our approach.

### 3.2.1 Critic Learning with HJB

The optimal value function is fully represented by the HJB equation. This PDE encodes the fundamental relationship between reward, system dynamics, and value evolution, and serves as the theoretical backbone for our value learning method. The following lemma states this connection formally in the multi-agent cooperative setting.

**Lemma 3.1** (HJB for Multi-agent Systems). For all $x \in \mathcal{X}$, the value function $V(x)$ is the optimal solution to satisfy the following HJB PDEs

$$-\rho V(x) + \nabla_x V(x)^\top f(x, u^*) + r(x, u^*) = 0, \tag{3}$$

where optimal control input $u^* = \arg\max_{u \in \mathcal{U}} \mathcal{H}(x, \nabla_x V(x))$. The Hamiltonian $\mathcal{H}$ is defined as $\mathcal{H} = \nabla_x V(x)^\top f(x, u) + r(x, u)$.

The proof is attached in Appendix A.1.

To approximate differentiable value functions, we solve the HJB PDEs in Eq. 3. Since conventional numerical methods become intractable beyond six state dimensions (Bui et al., 2022), we instead employ PINNs, which approximate value functions by minimizing PDE residuals. Specifically, we define the HJB PDE residuals as follows. Here, $u_t$ is the optimal control input in this paper.

$$\mathcal{R}_\theta(x_t) = -\rho V_\theta(x_t) + \nabla_{x_t} V_\theta(x_t)^\top f(x_t, u_t) + r(x_t, u_t). \tag{4}$$

and minimize the residual loss $\mathcal{L}_{\text{res}} = \left\| \mathcal{R}_\theta(x_t) \right\|_1$ towards zero during model refinement.

### 3.2.2 Policy Learning

While analytical optimal control laws can be derived in some cases by maximizing the Hamiltonian (Nakamura-Zimmerer et al., 2021; Shilova et al., 2024), such closed-form solutions are not available in complex multi-agent systems like MPE or MuJoCo. To overcome this challenge, we use an actor network to generate control inputs, replacing the need for analytical controls in the critic network used to approximate value functions. The actor and critic networks are refined iteratively until convergence, enabling the actor network to approximate optimal control policies. During training, we compute a continuous-time advantage function derived by residual $\mathcal{R}_\theta(x_t)$. This advantage function is used for policy gradient, where each agent's decentralized policy $\pi_{\phi_i}(u_i \mid x_t)$ is optimized to maximize long-term return.

**Lemma 3.2** (Instantaneous Advantage). Assume the one-step Q-function over a short interval $\delta t > 0$ be

$$Q(x_t, u_t) = r(x_t, u_t)\, \delta t + e^{-\rho \delta t} V(x_{t+\delta t}). \tag{5}$$

Then the instantaneous advantage satisfies

$$A(x_t, u_t) = -\rho V(x_t) + \nabla_{x_t} V(x_t)^\top f(x_t, u_t) + r(x_t, u_t). \tag{6}$$

The proof is attached in Appendix A.2.

With the critic's instantaneous advantage

$$A_\theta(x_t, u_t) = -\rho V_\theta(x_t) + \nabla_{x_t} V_\theta(x_t)^\top f(x_t, u_t) + r(x_t, u_t), \tag{7}$$

we update each agent's policy network $\pi_{\phi_i}(u_i \mid x_t)$ in a decentralized fashion. For agent $i$, we minimize the negative expected advantage under the joint policy

$$\mathcal{L}_{p_i} = -A_\theta(x_t, u_t) \log \pi_{\phi_i}(u_i \mid x_t), \tag{8}$$

Here, $u = (u_i, u_{-i})$ denotes the joint action, where $u_i \sim \pi_{\phi_i}$ is sampled from agent $i$'s policy, and $u_{-i}$ represents the actions of all other agents, sampled as $u_{-i} \sim \pi_{\phi_{-i}}$.

Since our actor is trained by using the advantage function, it is important to ensure that this update direction leads to policy improvement. The following lemma formalizes this property and shows that a gradient step on our actor loss yields a value-increasing policy update.

**Lemma 3.3** (Policy Improvement). Let $\pi_{\text{old}}$ be the current joint policy and $\pi_{\text{new}}$ the updated policy after one gradient step on the actor loss $\mathcal{L}_p$ with sufficiently small step size. Then

$$Q^{\pi_{\text{new}}}(x_t, u_t) \geq Q^{\pi_{\text{old}}}(x_t, u_t). \tag{9}$$

The proof can be found in Appendix A.3.

## 3.3 VALUE GRADIENT ITERATION MODULE

The performance of continuous-time control policies depends explicitly on the accuracy of the value, which in turn depends not only on the precision of its own approximation but also on the correctness of the value gradient $\nabla_x V(x)$. Recent studies have demonstrated that the accuracy of the value directly affects the learned control policies (Zhang et al., 2024b). However, enforcing the HJB PDE through residual loss alone does not guarantee accurate value *gradients*. Because in high-dimensional multi-agent systems, small gradient errors can be significantly amplified by coupled dynamics, leading to inaccurate policy gradients. To address this limitation, we introduce the VGI module, which explicitly propagates value gradients along sampled trajectories and refines the gradient estimates in a self-consistent manner. By iteratively updating $\nabla_x V(x)$ using the local dynamics and the current value approximation, VGI provides a trajectory-aligned correction signal that complements the global PDE constraint, resulting in substantially more accurate value-gradient learning.

**Definition 2** (VGI Gradient Estimator). Given a small time step $\Delta t$, the VGI estimator of the value gradient at $(x_t, u_t)$ is defined by

$$\nabla_{x_t} V(x_t) = \nabla_{x_t} r(x_t, u_t) \Delta t + e^{-\rho \Delta t} \nabla_{x_t} f(x_t, u_t)^\top \nabla_{x_{t+\Delta t}} V(x_{t+\Delta t}). \tag{10}$$

The VGI target in Eq. 10 can be interpreted as a one-step unrolling of the Bellman equations in the space of gradients. The first term captures the instantaneous contribution of the local reward gradient, while the second term propagates the downstream value information through the Jacobian of the system dynamics. This construction resembles a semi-discretized version of the value gradient flow and provides a practical surrogate for supervised gradient learning in the absence of ground-truth derivatives. The derivation process is posted at Appendix A.4. To justify why the VGI refinement is stable and converges to a promising gradient estimate, we analyze its update rule as a fixed-point iteration.

**Theorem 3.4** (Convergence of VGI). Let $G : \mathbb{R}^d \to \mathbb{R}^d$ be defined as

$$G(\zeta) = \nabla_{x_t} r(x_t, u_t) \Delta t + e^{-\rho \Delta t} \nabla_{x_t} f(x_t, u_t)^\top \zeta, \tag{11}$$

and assume the dynamics $\|\nabla_{x_t} f(x_t, u_t)\|$ is bounded . Then $G$ is a contraction, and the sequence $\zeta^{(k+1)} = G(\zeta^{(k)})$ converges to a unique fixed point $\zeta^* \in \mathbb{R}^d$.

The proof is detailed in Appendix A.5

Rather than introducing a separate network to predict value gradients, we directly compute the automatic derivative of the shared PINN critic $V_\theta(x_t)$. This gradient is then trained to match the VGI-generated target defined in Eq. 10. Specifically, we minimize the mean squared error between the computed and target gradients

$$\mathcal{L}_{\text{vgi}} = \left\| \nabla_{x_t} V_\theta(x_t) - \hat{g}_t \right\|_2^2, \tag{12}$$

where $\hat{g}_t = \nabla_{x_t} r_\phi(x_t, u_t) \Delta t + e^{-\rho \Delta t} \nabla_{x_t} f_\psi(x_t, u_t)^\top \nabla_{x_{t+\Delta t}} V_\theta(x_{t+\Delta t})$. Here, $r_\phi(x_t, u_t)$ denotes a reward model and $f_\psi(x_t, u_t)$ represents a dynamics model, where $\phi$ and $\psi$ are respective network parameters.

### 3.4 IMPLEMENTATION DETAILS

While the previous sections introduced our continuous-time actor–critic framework and the VGI module for value-gradient consistency, several practical considerations are essential to make the overall method operational and effective.

#### 3.4.1 DYNAMICS MODEL AND REWARD MODEL

In a continuous-time setting, the true dynamics are given by $\dot{x} = f(x, u)$, but directly learning $f$ via $\frac{x_{t+\Delta t} - x_t}{\Delta t}$ as a supervision target is pretty unstable in practice. Instead, we adopt a discrete-time model-based approach (Sutton, 1991; Hafner et al., 2019) that we train a neural network $f_\psi(x_t, u_t)$ to predict the next state $x_{t+\Delta t}$ via

$$\mathcal{L}_{\text{dyn}} = \left\| f_\psi(x_t, u_t) - x_{t+\Delta t} \right\|_2^2. \tag{13}$$

After learning $f_\psi$, we recover the continuous-time derivative by finite differences $\frac{f_\psi(x_t, u_t) - x_t}{\Delta t}$.

Similarly, we fit a reward network $r_\phi(x_t, u_t)$ to the observed instantaneous reward $r_t$

$$\mathcal{L}_{\text{rew}} = \left\| r_\phi(x_t, u_t) - r_t \right\|_2^2. \tag{14}$$

Both $f_\psi$ and $r_\phi$ are trained jointly, enabling us to compute the VGI module's target.

#### 3.4.2 ANCHOR LOSS FOR CRITIC NETWORK

In addition to the HJB residual loss, we incorporate a TD-style anchor loss to improve both the stability and accuracy of value learning. While the residual loss enforces the correctness of the value gradient, it does not constrain the value of $V(x)$. Terminal-condition losses can provide such supervision, but they often rely on access to well-defined terminal targets, which may be unavailable in complex continuous control environments such as MuJoCo. In these cases, the anchor loss offers an additional source of value landscape, helping the critic produce reasonable value approximations even when terminal rewards are sparse, delayed, or difficult to specify. We define the one-step return as

$$R_t = r(x_t, u_t) \Delta t + e^{-\rho \Delta t} V_\theta(x_{t+\Delta t}). \tag{15}$$

The anchor loss then enforces the value network to match these returns

$$\mathcal{L}_{\text{anchor}} = \left\| V_\theta(x_t) - R_t \right\|_2^2. \tag{16}$$

The overall critic objective combines all four losses

$$\mathcal{L}_{\text{total}} = \underbrace{\mathcal{L}_{\text{res}}}_{\text{HJB residual}} + \underbrace{\lambda_{\text{anchor}} \mathcal{L}_{\text{anchor}}}_{\text{TD anchor}} + \underbrace{\lambda_g \mathcal{L}_{\text{vgi}}}_{\text{VGI consistency}}. \tag{17}$$

Here $\lambda_{\text{anchor}}, \lambda_g$ are tunable weights balancing PDE residuals, value-bootstrap anchoring, and gradient consistency. In practice, we jointly train the reward model, dynamics model, and value network using the data from the current trajectory. The detailed training process is listed in Appendix A.6.

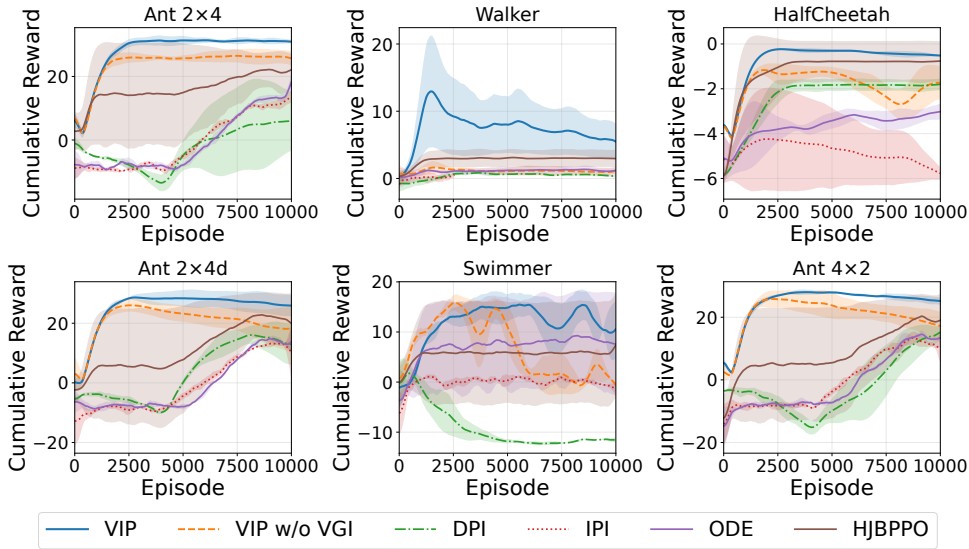

Figure 2: Performance across continuous-time multi-agent MuJoCo settings. The y-axis shows the mean cumulative reward.

# 4 EXPERIMENTAL RESULTS

We evaluate our **Value Iteration via PINN (VIP)** method on two continuous-time multi-agent benchmarks: MPE (Lowe et al., 2017) and multi-agent MuJoCo (Peng et al., 2021). In addition, we design a didactic benchmark, coupled oscillator (see details in Appendix B.1), to analyze value gradient approximations. This case study is easy to follow and enables the numerical computation of true values and their gradients, which provides a clear and interpretable setting to validate the effectiveness of VIP. Our experiments are designed to answer the following four key questions: **(1)** *Overall efficacy:* Does the proposed VIP model outperform existing continuous-time RL baselines in these environments? **(2)** *VGI ablation:* How much does the VGI module contribute to final performance and training stability? **(3)** *PINN design choice:* How does activation function choice and loss term weighting in the critic network affect the performance of VIP? **(4)** *Time discretization impact:* How well do discrete-time and continuous-time methods perform under arbitrary or unfixed time intervals? **(5)** *Comparison between VIP and discrete-time baselines:* Could discrete-time baselines perform comparable at continuous-time settings?

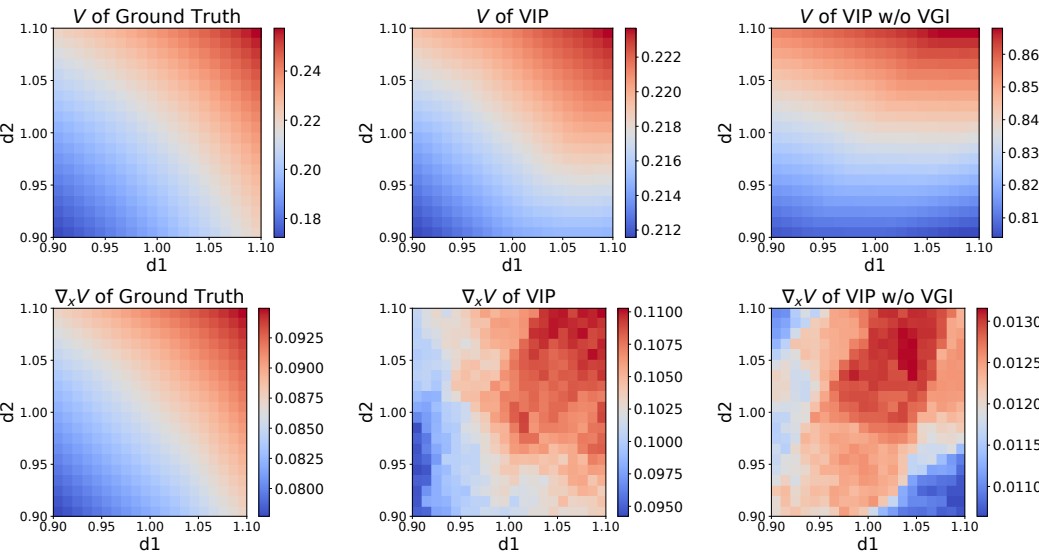

Figure 3: $V$ and $\nabla_x V$ contour using VIP w/ VGI and w/o VGI in $d_1$-$d_2$ frame.

**Benchmarks.** We evaluate our method against competitive baselines across eight experimental settings involving up to six agents and 113 state dimensions. We extend the MPE framework to a continuous-time formulation by using a variable-step Euler integration scheme, where the time interval $\Delta t$ is sampled from a predefined range at each step. Experiments are conducted on the *cooperative navigation* and *predator prey* environments. Similarly, we adapt MuJoCo to continuous-time settings and evaluate on a suite of multi-agent locomotion tasks, including *ant* ($2 \times 4$, $2 \times 4$d, $4 \times 2$), *walker*, *swimmer*, and *cheetah* ($6 \times 1$). Further implementation details of these benchmarks are provided in Appendix B.2 and B.3. Lastly, we introduce a simple yet illustrative `coupled oscillator` environment to highlight the behavior of exact value functions, value gradients, and the relative performance of different methods under a controlled setting.

## 4.1 BASELINE METHODS

To evaluate our CT-MARL framework VIP, we compare against four continuous-time policy iteration baselines and include an ablated variant of our method without VGI: **CT-MBRL (ODE)** (Yildiz et al., 2021): a continuous-time model-based RL approach that learns system dynamics via Bayesian neural ODEs. This method uses an actor-critic framework to approximate state-value functions and derive the continuous-time optimal policies.

**Differential Policy Iteration (DPI)** (Lee & Sutton, 2021): a model-based method with differential policy iterations that alternates between (i) solving the HJB PDEs to approximate continuous-time value functions and (ii) updating the policy by

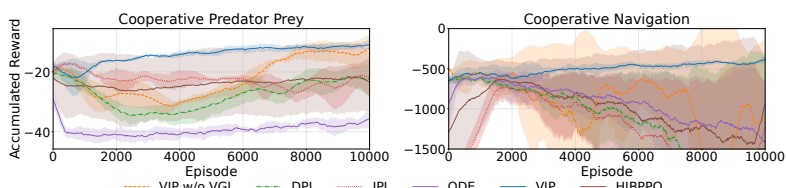

Figure 4: Performance across continuous-time MPE settings. The y-axis shows the mean cumulative reward.

following the instantaneous gradient of value approximations. **Integral Policy Iteration (IPI)** (Lee & Sutton, 2021): a partially model-free approach with integral policy iterations that reformulates the value function as a continuous integral, avoiding explicit differentiation during policy improvement. **HJBPPO** (Mukherjee & Liu): a recent method that employs a PINN-based critic to approximate the HJB residual and leverages a standard PPO-style policy optimization scheme to guide agent learning. In our experimental settings, we discretize the integral, roll out trajectories to accumulate rewards, and fit a policy to minimize the resultant value functions. **Ablation (w/o VGI)**: an ablated version of VIP without VGI, which isolates the efficacy of value gradient refinement. In addition, to evaluate how VIP compares with DTRL methods, we incorporate three widely used baselines in our experiments: MATD3 (Zhan et al., 2021), MAPPO (Yu et al., 2022), and MADDPG (Lowe et al., 2017).

## 4.2 RESULTS ANALYSIS

**Model Performance.** We evaluate all RL methods in MPE and MuJoCo environments using five random seeds and report the mean cumulative reward curves in Fig. 2 and 4. The results show that VIP with VGI consistently converges fastest and achieves the highest final return across all tasks, which empirically validates the efficacy of integrating PINNs with RL. As traditional time-dependent HJB equations are PDEs with a single boundary condition at terminal time, PINN may struggle to backpropagate the correct physics information when relying solely on boundary values, often resulting in poor value approximations (Krishnapriyan et al., 2021). This limitation becomes even more severe in the infinite-horizon setting, where the HJB formulation is time-independent and terminal losses are no longer available. To address this limitation, we incorporate the anchor and VGI loss terms in Eq. 17, which capture the landscape of value and its gradient so that PINNs converge to the true values. Ablation results further confirm the importance of VGI: removing VGI leads to significantly lower cumulative rewards across all experiments. This observation is consistent with the conclusion in the previous studies (Zhang et al., 2024a), which further strengthens that accurate value gradient approximation is crucial for effective PINN training and policy improvement. To further demonstrate the importance of VGI, we revisit the didactic example, generate 400 rollouts from sampled initial states, and compute the average value using models with and without VGI.

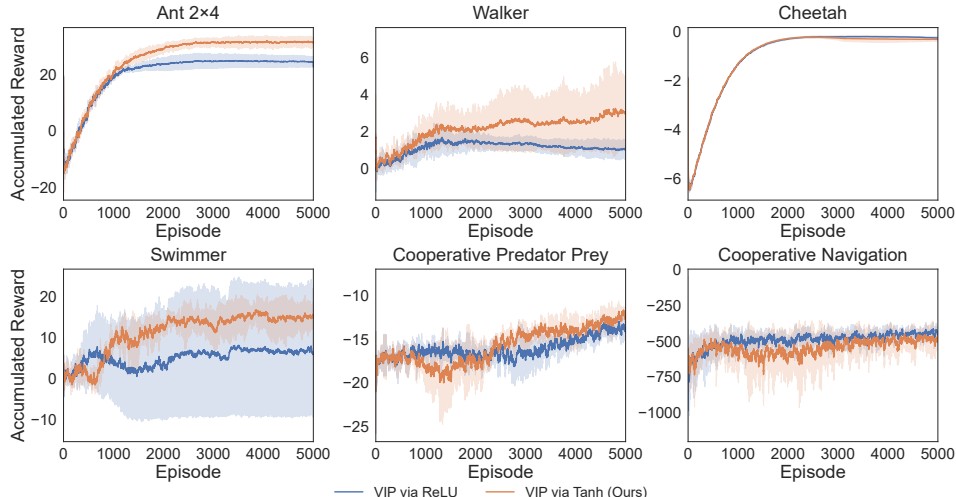

Figure 5: VIP performance with ReLU and Tanh activation functions in MuJoCo and MPE settings.

As shown in Fig. 3, the value contour projected onto the $d_1$–$d_2$ frame reveals that the model with VGI closely matches the ground truth in both structure and scale, while the model without VGI produces significantly biased approximations. These results confirm that adding VGI is necessary to improve the accuracy of value approximations. A detailed comparison of the corresponding value gradients is also illustrated in Fig. 3. Furthermore, we evaluate the sensitivity of each loss term for the critic network by measuring the minimum distance to prey in Fig. 6, which highlights the critical role of PINN in refining value and policy networks.

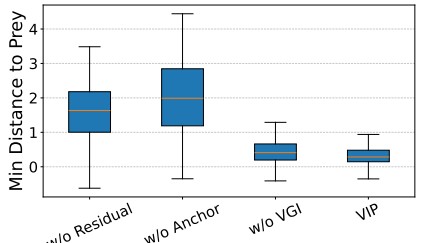

Figure 6: Ablation study of different loss terms in critic network for cooperative predator prey.

**Choice of PINN Design.** We first examine the impact of activation function choice when incorporating PINN-based losses into critic network refinement. Specifically, we train VIP using ReLU and Tanh activations in both MuJoCo and MPE environments and report the accumulated rewards in Fig. 5 and Fig. 11 (Appendix C.2). The results show that the VIP with Tanh consistently achieves higher accumulated rewards than the one with ReLU across all tasks. This experiment indicates the importance of activation function choice when using PINN-based losses to refine the critic network and has a consistent conclusion with the previous studies (Bansal & Tomlin, 2021; Zhang et al., 2024b). The good performance of Tanh can be attributed to its smoothness and differentiability, which are particularly important when using PINNs to solve PDEs.

PINNs often use fully-connected network architectures and rely on auto-differentiation to compute value gradients for PDEs. The PINN residuals, including PDEs, are further optimized using gradient descent. Smooth activation functions like Tanh support stable and

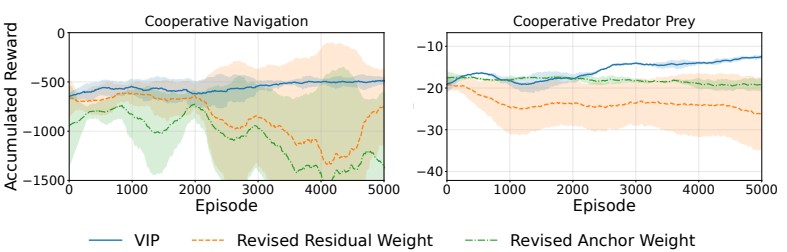

Figure 7: VIP performance with different weight settings in critic losses.

accurate gradient flow throughout training, enabling more effective value approximations. In contrast, VIP with ReLU often encounters zero-gradient regions during backpropagation, which results in gradient explosion for deeper network architectures or degrades the learning of value functions due to insufficient nonlinearity. Therefore, the choice of a smooth activation function like Tanh is better suited for physics-informed learning, thereby ensuring more accurate approx-

imations of the value functions. We also investigate the impact of weight parameters for PINN loss terms during VIP training. Specifically, we evaluate three configurations in Eq. 17: 1) balanced weights for all loss terms; 2) a large weight for anchor loss while keeping the others balanced; 3) a large weight for HJB residual with balanced weights for the remaining terms. As shown in Fig. 7, the best accumulated reward performance is achieved when all loss terms are properly balanced. Imbalanced weight settings yield stiffness in the training dynamics of PINNs (Wang et al., 2021), which makes the dominant loss term (with the largest weights) converge faster than the others (Wang et al., 2022). In our experiments, such an imbalance causes VIP only to satisfy the PDEs residuals or anchor loss during training, ultimately leading to poor value approximations.

**Time discretization impact.** Lastly, we evaluate the robustness of VIP and a well-trained MADDPG model (Lowe et al., 2017) by generating rollouts with varying time intervals in the didactic environment and computing the average return across these rollouts. Notably, in the original discrete-time training setting with a 4-dimensional state and 1-dimensional action space, MADDPG can already achieve near-optimal performance. Fig. 8 illustrates that VIP maintains a nearly constant return across different time intervals, whereas MADDPG's performance degrades significantly as the interval increases. This result highlights the advantage of VIP in continuous-time multi-agent scenarios. **DTRL algorithms' comparison.** Figure 9 compares VIP with three widely used discrete-

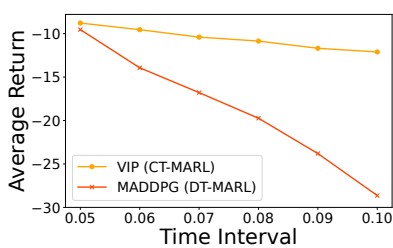

Figure 8: CT-MARL and DT-MARL performance under different time intervals.

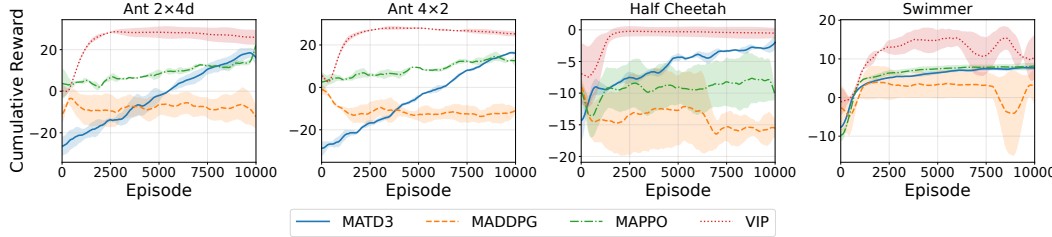

Figure 9: Comparison between VIP and DTRL baselines.

time MARL baselines (MATD3, MADDPG, MAPPO) across several multi-agent MuJoCo tasks. We observe that all discrete-time methods' performance degrades substantially in tasks such as Ant and HalfCheetah. This degradation is not surprising. Discrete-time Bellman backups implicitly assume fixed and sufficiently small time steps, and their value estimates become biased or unstable when the underlying system evolves in continuous-time with arbitrary or environment-dependent time intervals. Moreover, large integration steps amplify approximation errors in both the critic update and the target network, leading to inaccurate temporal-difference objective functions and unstable actor gradients, which aligns well with the observation in other works (Tallec et al., 2019; Park et al., 2021; De Asis & Sutton, 2024).

## 5 CONCLUSION

We propose a novel approach that integrates PINNs into the actor-critic framework to solve CT-MARL problems. Specifically, we approximate value functions using HJB-based PINNs and introduce VGI to improve value approximations, thus mitigating the adverse impact of inaccurate value approximations on policy learning. We validate our VIP across continuous-time variants of MPE and MuJoCo environments and empirically demonstrate that VIP converges faster and achieves higher accumulated reward compared to baselines of SOTA. Furthermore, we investigate the importance of activation function choice and loss term weighting for VIP performance. In summary, our proposed VIP offers a promising approach to solve CT-MARL problems. Since our current formulation is based on the HJB equation, we focus on cooperative settings; extending VIP to HJI-based formulations that handle competitive or mixed-motive scenarios is a compelling direction for future work.

ACKNOWLEDGMENTS

This work was supported by the National Science Foundation under Grant No. 2418106.

ETHICS STATEMENT

This work focuses on decision-making under continuous-time multi-agent systems. All experiments are conducted entirely in simulation and do not involve human subjects or personal data.

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

# A MATHEMATICAL PROOF

## A.1 PROOF OF LEMMA 3.1

*Proof.* Given all $x \in \mathcal{X}$ with a small horizon $\Delta t > 0$, we apply a Taylor expansion to the definition of $V(x)$ in Eq. 2 to derive the HJB PDE as follows

$$
\begin{aligned}
V(x) &= \max_{u \in \mathcal{U}} \int_t^\infty e^{-\rho(\tau - t)} r(x, u) d\tau \\
&= \max_{u \in \mathcal{U}} \int_t^{t+\Delta t} e^{-\rho(\tau-t)} r(x(\tau), u(\tau)) d\tau + \max_{u \in \mathcal{U}} \int_{t+\Delta t}^\infty e^{-\rho(\tau-t)} r(x(\tau), u(\tau)) d\tau \\
&= \max_{u \in \mathcal{U}} \int_t^{t+\Delta t} e^{-\rho(\tau-t)} r(x(\tau), u(\tau)) d\tau + \max_{u \in \mathcal{U}} \int_t^\infty e^{-\rho(s+\Delta t-t)} r(x(s+\Delta t), u(s+\Delta t)) ds \\
&= \max_{u \in \mathcal{U}} \int_t^{t+\Delta t} e^{-\rho(\tau-t)} r(x(\tau), u(\tau)) d\tau + e^{-\rho \Delta t} \max_{u \in \mathcal{U}} \int_t^\infty e^{-\rho(s-t)} r(x(s+\Delta t), u(s+\Delta t)) ds \\
&\approx \max_{u \in \mathcal{U}} \int_t^{t+\Delta t} e^{-\rho(\tau-t)} r(x(\tau), u(\tau)) d\tau + e^{-\rho \Delta t} \max_{u \in \mathcal{U}} \int_{t+\Delta t}^\infty e^{-\rho(\tau-t)} r(x(\tau), u(\tau)) d\tau \\
&= \max_{u \in \mathcal{U}} r(x, u) \Delta t + e^{-\rho \Delta t} \max_{u \in \mathcal{U}} V(x(t+\Delta t)) \\
&= \max_{u \in \mathcal{U}} r(x, u) \Delta t + (1 - \rho \Delta t + o(\Delta t)) \max_{u \in \mathcal{U}} (V(x) + \nabla_x V(x)^\top f(x, u) \Delta t + o(\Delta t))
\end{aligned}
$$

By canceling out $V(x)$ on both sides of the above equality, we obtain that

$$
-\rho V(x) \Delta t + \max_{u \in \mathcal{U}} (\nabla_x V(x)^\top f(x, u) + r(x, u)) \Delta t + o(\Delta t) = 0
$$

Dividing by $\Delta t$ and letting $\Delta t \to 0$, we have that

$$
-\rho V(x) + \max_{u \in \mathcal{U}} (\nabla_x V(x)^\top f(x, u) + r(x, u)) = 0.
$$

Therefore, the $V(x)$ is the optimal solution to the following HJB PDEs:

$$
-\rho V(x) + \nabla_x V(x)^\top f(x, u^*) + r(x, u^*) = 0,
$$

Here optimal control input is $u^* = \arg\max_{u \in \mathcal{U}} \mathcal{H}(x, \nabla_x V(x))$, where $\mathcal{H}$ is the Hamiltonian defined as $\mathcal{H} = \nabla_x V(x)^\top f(x, u) + r(x, u)$. $\qquad\square$

## A.2 PROOF OF LEMMA 3.2

*Proof.* Recall the one-step $Q$-function over a short interval $\delta t > 0$, where $u$ is the optimal control input

$$
Q(x_t, u_t) = r(x_t, u_t) \delta t + e^{-\rho \delta t} V(x_{t+\delta t}).
$$

For small $\delta t$ we have the first-order Taylor expansion in state:

$$
V(x_{t+\delta t}) = V(x_t) + \nabla_{x_t} V(x_t)^\top f(x_t, u_t) \delta t + o(\delta t).
$$

Similarly, $e^{-\rho \delta t} = 1 - \rho \delta t + o(\delta t)$.

Plugging both expansions into $Q(x, u)$ gives

$$
Q(x_t, u_t) = r(x_t, u_t) \delta t + (1 - \rho \delta t + o(\delta t)) \left[ V(x_t) + \nabla_{x_t} V(x_t)^\top f(x_t, u_t) \delta t + o(\delta t) \right]
$$

$$
= r(x_t, u_t) \delta t + V(x_t) + \left[ \nabla_{x_t} V(x_t)^\top f(x_t, u_t) - \rho V(x_t) \right] \delta t + o(\delta t).
$$

Subtract $V(x_t)$ and discard the higher-order term:

$$
Q(x_t, u_t) - V(x_t) = [-\rho V(x_t) + \nabla_{x_t} V(x_t)^\top f(x_t, u_t) + r(x_t, u_t)] \delta t + o(\delta t).
$$

Dividing by $\delta t$ and letting $\delta t \to 0$ yields the instantaneous advantage density

$$
A(x_t, u_t) = \lim_{\delta t \to 0} \frac{Q(x_t, u_t) - V(x_t)}{\delta t} = -\rho V(x_t) + \nabla_{x_t} V(x_t)^\top f(x_t, u_t) + r(x_t, u_t).
$$

This completes the proof. $\qquad\square$

## A.3 PROOF OF LEMMA 3.3

*Policy Improvement via State-Action Value Function.* We consider the standard policy improvement step, where the new policy is obtained by maximizing the state-action value function

$$\pi_{\text{new}}(x_t) = \arg\max_{u \in \mathcal{U}} Q^{\pi_{\text{old}}}(x_t, u_t),$$

where the one-step Q-function with a small $\delta t > 0$ is defined as:

$$Q(x_t, u_t) = r(x_t, u_t)\,\delta t + e^{-\rho \delta t} V(x_{t+\delta t})$$
$$= r(x_t, u_t)\,\delta t + e^{-\rho \delta t} \mathbb{E}_{u' \sim \pi(\cdot | x')} \left[ Q(x'_{t+\delta t}, u'_{t+\delta t}) \right],$$

Since we assume the goal state function stays invariant, we only define the $Q^\pi = \int_0^T e^{-\rho t} r_t dt$. Because the new policy $\pi^{\text{new}}$ yields equal or higher value in expectation

$$\mathbb{E}_{u \sim \pi_{\text{new}}} \left[ Q^{\pi_{\text{old}}}(x_t, u_t) \right] \geq \mathbb{E}_{u \sim \pi_{\text{old}}} \left[ Q^{\pi_{\text{old}}}(x_t, u_t) \right].$$

Then we can have

$$Q^{\pi_{\text{old}}} = r_{t_0}\,\delta t + e^{-\rho \delta t} (\mathbb{E}_{u_{t_1} \sim \pi_{\text{old}}} [Q^{\pi_{\text{old}}}(x_{t_1}, u_{t_1})])$$
$$\leq r_{t_0}\,\delta t + e^{-\rho \delta t} \left( \mathbb{E}_{u_{t_1} \sim \pi_{\text{new}}} [Q^{\pi_{\text{old}}}(x_{t_1}, u_{t_1})] \right)$$
$$= r_{t_0}\,\delta t + e^{-\rho t_1} r_{t_1}\,\delta t + e^{-\rho \delta t} \left( \mathbb{E}_{u_2 \sim \pi_{\text{old}}} [Q^{\pi_{\text{old}}}(x_{t_2}, u_{t_2})] \right)$$
$$\leq r_{t_0}\,\delta t + e^{-\rho t_1} r_{t_1}\,\delta t + e^{-\rho \delta t} \left( \mathbb{E}_{u_{t_2} \sim \pi_{\text{new}}} [Q^{\pi_{\text{old}}}(x_{t_2}, u_{t_2})] \right)$$
$$= r_{t_0}\,\delta t + e^{-\rho t_1} r_{t_1}\,\delta t + e^{-\rho t_2} r_{t_2}\,\delta t + e^{-\rho \delta t} \left( \mathbb{E}_{u_{t_3} \sim \pi_{\text{old}}} [Q^{\pi_{\text{old}}}(x_{t_3}, u_{t_3})] \right).$$
$$\vdots$$
$$\leq \int_t^\infty e^{-\rho t} r_t dt$$
$$= Q^{\pi_{\text{new}}}.$$

$\square$

## A.4 DERIVATION OF DEFINITION 2

*Proof.* We consider the value function defined in Eq. 2 and follow the Proof of Lemma 3.1 to write out the dynamic programming principle of $V(x_t)$ as

$$V(x_t) = r(x_t, u_t)\Delta t + e^{-\rho \Delta t} V(x_{t+\Delta t}).$$

where $u_t$ is the optimal control input. Taking the gradient with respect to $x$ on both sides using the chain rule

$$\nabla_{x_t} V(x_t) = \nabla_{x_t} r(x_t, u_t)\Delta t + e^{-\rho \Delta t} \nabla_{x_t} f(x_t, u_t)^\top \nabla_{x_{t+\Delta t}} V(x_{t+\Delta t}).$$

which matches the estimator proposed in Definition 2. $\square$

## A.5 PROOF OF THEOREM 3.4

*Proof.* From the definition of VGI in Definition 2, applying a first-order Euler step gives

$$x_{t+\Delta t} = x_t + f(x_t, u_t)\Delta t + o(\Delta t).$$

This yields the first-order VGI approximation

$$\nabla_{x_t} V(x_t) = \nabla_{x_t} r(x_t, u_t)\Delta t + e^{-\rho \Delta t} \left[ I + \nabla_{x_t} f(x_t, u_t)\Delta t \right]^\top \nabla_{x_{t+\Delta t}} V(x_{t+\Delta t}).$$

We can rewrite this approximation as an affine map

$$\zeta = G(\zeta)$$

where

$$b = \nabla_{x_t} r(x_t, u_t)\Delta t, \ A = e^{-\rho \Delta t} \left[ I + \nabla_{x_t} f(x_t, u_t)\Delta t \right]^\top, \ \zeta = \nabla_x V(x).$$

In Section 3.1, the joint dynamics function $f(x, u)$ is assumed time-invariant, which implies its Jacobian $\nabla_x f$ is also time-independent. Consequently, the matrix $A$ is time-invariant, and the update map can be written as a single contraction map

$$G(\zeta) = b + \mathcal{A}\zeta,$$

rather than a family of time-indexed maps $G_t$. Assuming the dynamics have a bounded Jacobian,

$$\|\nabla_x f(x_t, u_t)\| \le L_f,$$

we obtain the bound

$$\|A\| \le e^{-\rho \Delta t}(1 + L_f \Delta t) = \beta.$$

Because we study high-frequency settings, $\Delta t$ is chosen sufficiently small, which makes $L_f \Delta t \to 0$, so that $\beta < 1$. For any $\zeta_1, \zeta_2 \in \mathbb{R}^d$,

$$\|G(\zeta_1) - G(\zeta_2)\| = \|A(\zeta_1 - \zeta_2)\| \le \beta\|\zeta_1 - \zeta_2\|,$$

Hence, $G$ is a contraction. Banach's fixed-point theorem guarantees a unique fixed point $\zeta^*$ and linear convergence $\|\zeta^{(k)} - \zeta^*\| \le \beta^k \|\zeta^{(0)} - \zeta^*\|$. Therefore, the value-gradient iteration converges, completing the proof. □

### A.6 TRAINING ALGORITHM

We present the training algorithm for our proposed approach, Value Iteration via PINN (VIP), as follows

---

**Algorithm 1** Value Iteration via PINN (VIP)

---

1: **Init:** value net $V_\theta$, policy nets $\{\pi_{\omega_i}\}_{i=1}^N$, dynamics $\hat{f}_\psi$, reward $\hat{r}_\phi$
2: **for** $l = 1, \ldots, T$ **do**
3:     ▷ **Collect one rollout:**
4:     $x \leftarrow \text{env.reset}()$
5:     **for** $k = 1, \ldots, K$ **do**
6:         sample decision time $t \sim \mathcal{T}$                            ▷ $t$ is arbitrary time
7:         **for** each agent $i = 1, \ldots, N$ **do**
8:             $u_i \sim \pi_{\omega_i}(u_i \mid x)$
9:         **end for**
10:        set joint action $u = (u_1, \ldots, u_N)$
11:        $(x', r) \leftarrow \text{env.step}(u)$
12:        append $(x, u, r, x')$ to local rollout $\mathcal{R}$
13:        $x \leftarrow x'$
14:     **end for**
15:     ▷ **Dynamics and Reward Model learning on $\mathcal{R}$**
16:     update $\psi, \phi$ as per the Eq. 13 and 14.
17:     ▷ **Critic update on $\mathcal{R}$**
18:     compute all losses $\mathcal{L}_{\text{res}}, \mathcal{L}_{\text{anchor}}, \mathcal{L}_{\text{vgi}}$ by Eq. 17.
19:     $\theta \leftarrow \theta - \alpha_V \nabla_\theta(\ldots)$
20:     ▷ **Actor update for each agent**
21:     **for** $i = 1, \ldots, N$ **do**
22:        compute $A(x, u)$ for all $(x, u) \in \mathcal{R}$
23:        $\omega_i \leftarrow \omega_i - \alpha_\pi \nabla_{\omega_i}\big(-\mathbb{E}_{(x,u)\in\mathcal{R}}[A(x, u) \log \pi_{\omega_i}(u_i \mid x)]\big)$ by Eq. 8.
24:     **end for**
25: **end for**

---

## B ENVIRONMENTAL SETTINGS

### B.1 COUPLED OSCILLATOR

We evaluate on a two-agent *coupled spring–damper* system. Each agent $i \in \{1, 2\}$ controls one mass in a pair of identical oscillators with linear coupling. The continuous-time dynamics are

$$\begin{aligned} \dot{x}_i &= v_i, \\ \dot{v}_i &= -k\,x_i \,-\, b\,v_i \,+\, u_i, \end{aligned} \qquad i = 1, 2, \tag{18}$$

where

- $x_i$ and $v_i$ are the position and velocity of mass $i$;
- $k = 1.0$ is the spring constant, and $b = 0.5$ is the damping coefficient;
- $u_i \in [-u_{\max}, u_{\max}]$ is the control force applied by agent $i$, with $u_{\max} = 10$.

At each step the joint reward is

$$r = -\Big(x_1^2 + x_2^2 \; + \; \lambda_c\,(x_1 - x_2)^2 \; + \; \beta\,(u_1^2 + u_2^2)\Big),$$

with coupling strength $\lambda_c = 2.0$ and control penalty $\beta = 0.01$. We normalize by a constant factor (here $1/10$) so that $r \in [-1, 0]$.

For the coupled oscillator with linear dynamics

$$\dot{x} = A\,x + B\,u,$$

We can compute the exact infinite-horizon LQR solution

1. Solve the continuous algebraic Riccati equation (CARE)
$$A^\top P + P\,A - P\,B\,R^{-1}B^\top P + Q = 0$$
for the symmetric matrix $P \in \mathcal{R}^{4\times 4}$.

2. Form the optimal state-feedback gain
$$K \;=\; R^{-1}B^\top P.$$

3. The optimal control law is
$$u^*(x) \;=\; -\,K\,x, \qquad u_i^* = -K_i\,x,$$
where $K_i$ is the $i$-th row of $K$.

4. The corresponding optimal value function is the quadratic form
$$V^*(x) \;=\; x^\top P\,x,$$
whose exact gradient is
$$\nabla_x V^*(x) \;=\; 2\,P\,x.$$

We use $u^*(x)$, $V^*(x)$, and $\nabla_x V^*(x)$ as ground truth targets when evaluating the precision of the policy, the error of the value function, and the consistency of the gradient.

## B.2 CONTINUOUS-TIME MPE

We build on the standard MPE of Lowe et al. (2017), which simulates $N$ holonomic agents in a 2D world with simple pairwise interactions. In the original MPE each control step advances the physics by a fixed time-step $\Delta t_{\text{fixed}} = 0.1\,\text{s}$. To evaluate our continuous-time framework under irregular sampling, we modify the simulator so that at each step the integration interval is drawn randomly,

$$\Delta t_k \;\sim\; \text{Uniform}(\Delta t_{\min}, \Delta t_{\max}).$$

The underlying dynamics, observation and action spaces, reward functions, and task definitions remain exactly as in the original MPE. For the cooperative predator-prey environment, we only control the predators (3 agents) action to capture the prey (1 agent). While the prey is set up with random actions.

## B.3 CONTINUOUS-TIME MULTI-AGENT MUJOCO

For high-dimensional control we adapt the discrete-time Multi-Agent MuJoCo suite (e.g. cooperative locomotion, quadruped rendezvous). By default MuJoCo uses an internal physics integrator with a base time-step of $0.01\,\text{s}$ and repeats each action for $K_{\text{fixed}} = 5$ frames, yielding an effective control interval $\Delta t_{\text{fixed}} = 0.05\,\text{s}$. We instead sample the number of frames per control step,

$$K_k \;\sim\; \text{Uniform Integer}(1, 9),$$

so that each step advances by

$$\Delta t_k = K_k \times 0.01\,\text{s} \;\in\; [0.01,\, 0.09]\text{s}$$

at random. All other aspects of the environment (observations, reward structure, termination conditions) are kept identical to the original multi-agent MuJoCo tasks.

## C  ADDITIONAL EXPERIMENTAL RESULTS

Experiments were conducted on hardware comprising an Intel(R) Xeon(R) Gold 6254 CPU @ 3.10GHz and four NVIDIA A5000 GPUs. This setup ensures the computational efficiency and precision required for the demanding simulations involved in multi-agent reinforcement learning and safety evaluations.

### C.1  VALUE GRADIENT COMPARISON

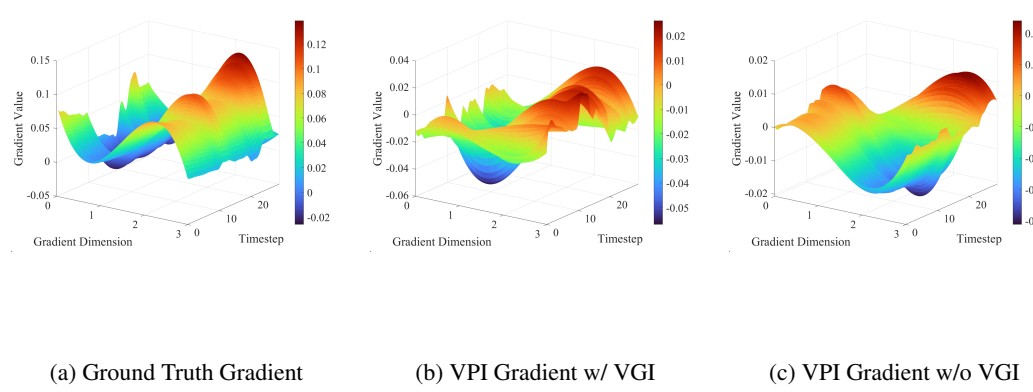

(a) Ground Truth Gradient   (b) VPI Gradient w/ VGI   (c) VPI Gradient w/o VGI

Figure 10: Value gradient comparison between using of VPI module.

To further demonstrate the effectiveness of VIP, we sample the same trajectory in the coupled oscillator environment and compute the value gradient from the analytical LQR solution, from VIP equipped with the VGI module (ours), and from VIP trained without VGI, respectively. Fig. 10 presents the resulting 3-D surfaces. The surface in panel (b) preserves the principal ridges and valleys of the ground truth, showing that the network recovers the correct geometric structure of $\nabla_x V$; its absolute error remains below $0.02$ across almost all timesteps and gradient dimensions. In contrast, the surface in panel (c) is noticeably distorted: several peaks are flattened, troughs are misplaced, and the absolute error frequently exceeds $0.08$. This comparison confirms that the VIP module is critical for aligning the learned gradients with the analytical solution, thereby reducing bias and stabilising the HJB residual.

### C.2  RELU VS TANH AT ANT $2 \times 4d$ AND ANT $4 \times 2$.

Fig. 11 compares VIP's learning curves when the policy network uses ReLU or Tanh activations on Ant $2 \times 4d$ and Ant $4 \times 2$. Across both tasks the Tanh implementation converges faster and

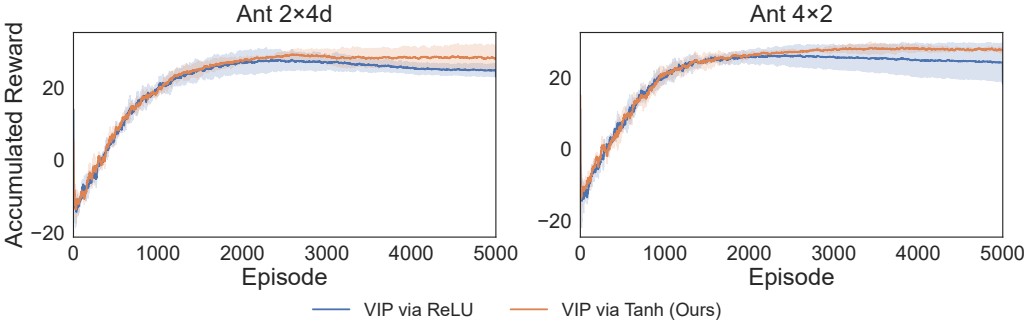

Figure 11: VIP performance with ReLU and Tanh at Ant $2 \times 4d$ and Ant $4 \times 2$.

attains a higher plateau reward, whereas the ReLU version peaks earlier and then undergoes a mild performance decay. The observation aligns with the earlier Tanh-versus-ReLU analysis reported in the main paper Fig. 5: smoother activation functions mitigate gradient saturation and promote more stable policy updates. The additional evidence from the two Ant variants therefore reinforces our previous claim that Tanh is better suited for value–gradient propagation in VIP.

## C.3 STOCHASTIC NOISE & MODEL MISMATCH

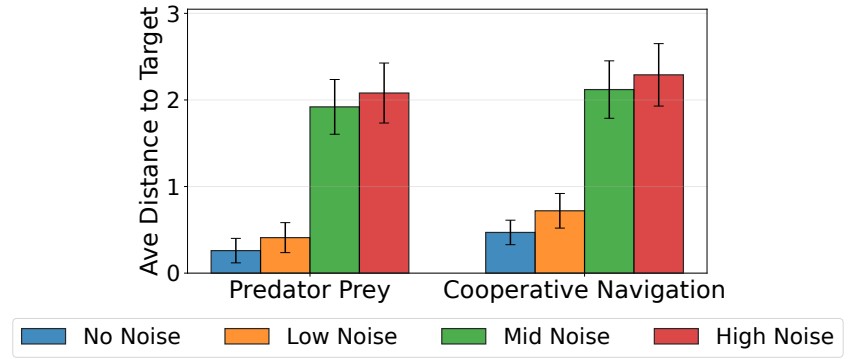

Figure 12: Performance under different noise levels.

Figure 12 evaluates the robustness of VIP under different levels of stochastic perturbation injected directly into the dynamics, according to

$$x_{t+\Delta t} = f(x_t, u_t)\,\Delta t + \varepsilon_t, \quad \varepsilon_t \sim \mathcal{N}(0, \sigma^2 I),$$

where we consider three noise scales: $\sigma^2 = 0.1$ (Low), $0.5$ (Mid), and $1.0$ (High). With no or low noise, VIP maintains strong performance across both Predator–Prey and Cooperative Navigation, indicating that the learned continuous-time value function and its gradients remain stable under mild stochasticity. As the noise level increases, performance degrades noticeably, which is expected: our method is designed for deterministic continuous-time systems, and stochastic settings fundamentally alter the associated HJB equations (leading to stochastic HJB or HJB–Bellman–Fokker–Planck forms). Since VIP does not incorporate additional components for modeling diffusion terms or uncertainty, the degradation at high noise levels is consistent with the theoretical scope of our formulation.

## C.4 SENSITIVITY CHECK FOR $\Delta t$

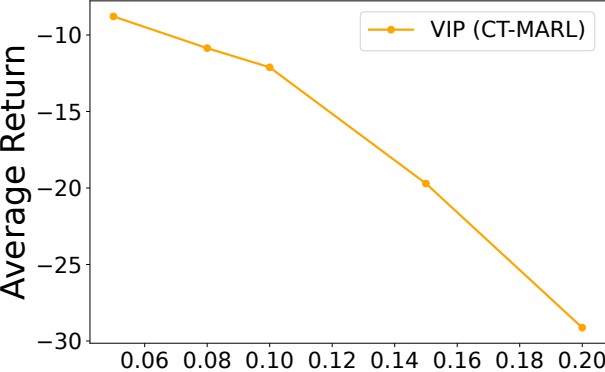

Figure 13: Average return under different $\Delta t$ in coupled oscillator .

For sufficiently small step sizes, VIP remains stable and achieves strong return, consistent with the contraction guarantee. As $\Delta t$ increases, the first-order discretization of the continuous-time dynamics

becomes less accurate, and the local linearization used in VGI and PDE no longer matches the true state evolution. Once $\Delta t$ exceeds a task-dependent threshold (e.g., $\Delta t \approx 0.15$ in the Coupled Oscillator), approximation errors accumulate across iterations and the performance begins to degrade. For very large $\Delta t$, the contraction conditions are effectively violated and the value-gradient update can no longer maintain stability, leading to a sharp drop in performance. This behavior matches our theoretical interpretation of VGI: the bounded-Jacobian and small–$\Delta t$ requirements are sufficient conditions for contraction, the method remains robust over a wide range of practical step sizes, but the performance will degrade sharply if the $\Delta t$ far from the effective region of $\Delta t$.

Table 1: Hyperparameter settings used in all experiments.

| Parameter | Value |
|---|---|
| Episode length | 50 |
| Replay buffer size | $10^4$ |
| Discount factor $\rho$ | 0.95 |
| Soft update rate $\tau$ | 0.001 |
| Actor learning rate | 0.0001 |
| Critic learning rate | 0.001 |
| Dynamics model learning rate | 0.001 |
| Reward model learning rate | 0.001 |
| Exploration steps | 1000 |
| Model save interval | 1000 |
| Random seed | 111-120 |

## D  HYPER-PARAMETERS

As Table 1 shows, the *exploration steps* are used to delay the decay of the exploration rate: during the first 1000 steps, the exploration schedule remains fixed to encourage initial exploration. The *soft update rate $\tau$* controls the target network update in the critic and value estimation, where we adopt a target network with an exponential moving average to stabilize bootstrapped training. This technique helps suppress oscillations in value learning and leads to more accurate estimation of long-horizon returns.

Table 2: Summary of neural network architectures used in our framework.

| Network | Input Dimension | Architecture and Activation |
|---|---|---|
| **Value Network** | State $(d)$ | FC(128) $\rightarrow$ FC(128) $\rightarrow$ FC(1), ReLU or Tanh |
| **Dynamics Network** | State + Joint Action $(d + na)$ | FC(128) $\rightarrow$ FC(128) $\rightarrow$ FC($d$), ReLU |
| **Reward Network** | State + Joint Action $(d + na)$ | FC(128) $\rightarrow$ FC(128) $\rightarrow$ FC(1), ReLU |
| **PolicyNet** | Observation + Time Interval $(o + 1)$ | FC(128) $\rightarrow$ FC(128) $\rightarrow$ FC(64) $\rightarrow$ FC($a$), ReLU |

Table 2 summarizes the architectures of the neural networks used in our VIP framework. All networks are implemented as FC layers with hidden size 128 unless otherwise specified. The value network takes the concatenation of the global state and time $(d)$ as input and outputs a scalar value. In our implementation, we use the **Tanh** activation function for the value network, as it provides smoother and more stable gradient propagation, which is critical for PINN-based value approximation. To validate this choice, we conducted an ablation study comparing **Tanh** and **ReLU** activations in the previous section.

## E  THE USE OF LARGE LANGUAGE MODELS (LLMS)

We use LLMs as a writing assistant to polish/revise the paper.

