# OpenReview forum: "Continuous-Time Value Iteration for Multi-Agent Reinforcement Learning"
_ICLR.cc/2026/Conference — ICLR 2026 Poster_

### Official Review · Reviewer_b9mm · 2025-10-21

**Soundness:** 3
**Presentation:** 3
**Contribution:** 3
**Rating:** 6
**Confidence:** 2

**Summary:**

This paper proposes a continuous-time multi-agent reinforcement learning (CT-MARL) framework called VIP, which leverages PINNs to approximate value functions as solutions to HJB.
To address inaccuracies in value gradients that can destabilize policy learning in multi-agent settings, the authors introduce a Value Gradient Iteration (VGI) module that iteratively refines gradients along trajectories, improving value approximation and overall performance.
Overall, the paper extends continuous-time RL to multi-agent cooperative scenarios, demonstrating good performance over baselines while mitigating the curse of dimensionality.

**Strengths:**

- The integration of PINNs with a VGI module is a fresh approach to CT-MARL, addressing key challenges like the curse of dimensionality and gradient inaccuracies in high-dimensional multi-agent systems.

- Provides solid mathematical foundations, including lemmas and theorems with proofs.

- Demonstrates genuine advantage of CT training under varying $\Delta t$ and nice stress test against a competitive DT agent.

**Weaknesses:**

- VGI targets rely on accurate dynamics and reward models, if they have a little difference, the estimation may differs a lot and fails.

- VGI convergence relies on bounded Jacobians and small $\Delta t$; it’s not fully clear how sensitive VIP is when these conditions are loosened

- Discrete-time MARL baselines are excluded from the main comparisons.

**Questions:**

- How sensitive is VGI to bias/variance in dynamics and rewards? Have you tried perturbing these models or training with deliberately misspecified dynamics to see VIP’s robustness envelope?

- How does VIP behave with very small/large variable $\Delta t$? Any adaptive-step control in the algorithm?

---

> ### Author Response · Authors · 2025-11-24
> **We thank the reviewer for the valuable comments, we provide detailed responses below.**
>
> Many thanks to the reviewer's careful reading of our work and the valuable points raised. We provide one-by-one direct response below:
>
> **W1. VGI targets rely on accurate dynamics and reward models, if they have a little difference, the estimation may differ a lot and fails**
>
> We agree that VGI relies on reasonably accurate dynamics and reward models, as it is a model-based method. To quantify this sensitivity, we performed controlled experiments with injected stochastic disturbances (Figure 14, Appendix C.5). In the low-variance setting ($\sigma^2 = 0.1$), performance remains largely stable, indicating that the method tolerates small model deviations. However, when the noise level increases to $\sigma^2 = 0.5$ and $\sigma^2 = 1.0$, performance drops sharply.
>
> This behavior is expected: our HJB formulation is deterministic, and VGI assumes smooth, well-defined value gradients derived from deterministic dynamics. Large stochasticity violates these assumptions, leading to biased PDE residuals and inconsistent value-gradient approximations.
>
> **W2.  VGI convergence relies on bounded Jacobians and small $\Delta t$; it’s not fully clear how sensitive VIP is when these conditions are loosened**
>
> We conducted performance experiments of VIP at different time steps in Figure 15 (Appendix C.6), and the results show that VIP remains stable and achieves consistent performance at smaller time steps. When $\Delta t$ becomes larger, the approximated error accumulates, and the VIP deviates from its theoretical contraction behavior.
>
> **W3. Discrete-time MARL baselines are excluded from the main comparisons.**
>
> We agree that adding more discrete-time baselines will help readers understand better. We have added three baselines as MADDPG, MAPPO, MATD3. Our results have been displayed in Figure 9, which is marked blue in the experimental part. All discrete-time methods' performance degrades in these tasks. But this degradation is expected: discrete-time Bellman backups implicitly assume fixed time steps, and their value approximations become inaccurate when the underlying system evolves in continuous time with arbitrary time intervals.
>
> For completeness, we also evaluate our algorithms in discrete-time settings in Figure 13 (Appendix C.4). The experimental results still show that our algorithms perform comparably with these discrete-time baselines. Importantly, the previous performance gains observed in our main experiments diminish under fixed-step discrete-time dynamics, indicating that our advantages stem from modeling continuous-time evolution directly, rather than from implementation choices or evaluation bias.
>
> **Q1. How sensitive is VGI to bias/variance in dynamics and rewards?**
>
> To verify the impact of the model mismatch, we inject noise into the dynamics model as $x_{t+\Delta t} = f(x_t, u_t)\Delta t + \varepsilon_t, \varepsilon_t \sim \mathcal{N}(0,\sigma^2 I)$. We set up the noise levels as $\sigma^2 = 0.1$ (low noise), $\sigma^2 = 0.5$ (mid noise) and $\sigma^2 = 1$ (high noise). With the low variance setting ($\sigma^2=0.1$), the performance does not degrade too much, which means that our algorithm can still tolerate small deviations. However, when the noise level increases to $\sigma^2=0.5$ and $\sigma^2=1.0$, the performance degraded terribly. This result is consistent with our expectation: the underlying HJB formulation is deterministic rather than stochastic, and large stochasticity breaks the assumptions required for consistent value-gradient learning.
>
> **Q2. How does VIP behave with very small/large variable $\Delta t$? Any adaptive-step control in the algorithm?**
>
> As we discussed in W2 and Figure 15, VIP remains stable and achieves consistent performance over small step sizes. When $\Delta t$ increases too much, the approximation error accumulates, and the VIP deviates from its theoretical contraction behavior.
>
> Regarding the adaptive-step control, we would like to clarify that in continuous-time MARL, $\Delta t$ is not an optimization variable. Instead, it is part of the environment dynamics: a non-fixed simulation time resolution used to approximate the continuous-time system. The agent does not choose $\Delta t$, but after training, it can learn to adapt to a variable $\Delta t$, which aligns well with our motivation.

---

### Official Review · Reviewer_qN3L · 2025-10-23

**Soundness:** 2
**Presentation:** 2
**Contribution:** 2
**Rating:** 2
**Confidence:** 5

**Summary:**

This paper proposes a **Continuous-Time Multi-Agent Reinforcement Learning (CT-MARL)** framework, termed **VIP (Value Iteration via PINN)**, which formulates MARL in the continuous-time domain.
The authors argue that conventional discrete-time RL (DTRL) methods struggle when applied to systems requiring high-frequency decision-making or irregular time intervals. To address this, VIP replaces discrete Bellman recursion with differentiable value functions that satisfy the **Hamilton–Jacobi–Bellman (HJB)** equations.

Key contributions include:

1. Leveraging **Physics-Informed Neural Networks (PINNs)** to approximate differential value functions governed by HJB PDEs.
2. Introducing a **Value Gradient Iteration (VGI)** module to refine gradient estimation and stabilize value learning.
3. Evaluating VIP on continuous-time versions of MPE and multi-agent MuJoCo environments, demonstrating improved sample efficiency and performance over existing continuous-time baselines (e.g., ODE-based RL, HJBPPO, DPI, and IPI).

**Strengths:**

* **Timely and ambitious topic:**
  The paper tackles the underexplored area of continuous-time MARL, aiming to bridge PDE-based control theory and deep RL.

* **Interesting methodological combination:**
  Integrating PINNs into an actor–critic framework and adding a value-gradient refinement mechanism (VGI) reflects a creative blend of ideas from model-based RL and physics-informed learning.

* **Comprehensive derivations:**
  Theoretical background and mathematical formulations (e.g., derivations of HJB, VGI, convergence proofs) are detailed and rigorous.

* **Empirical effort:**
  Experiments on both MPE and multi-agent MuJoCo show that VIP achieves improved training stability and faster convergence than existing continuous-time baselines.

* **Potential significance:**
  The exploration of continuous-time MARL could inspire future research directions that better align RL with real-world physical systems and continuous control.

**Weaknesses:**

Despite its conceptual ambition, the paper faces several **foundational and empirical limitations**, which prevent it from reaching ICLR acceptance standards.

### (1) Motivation and conceptual clarity

The motivation for formulating MARL in continuous time is **not well-justified**.
While the authors mention irregular time intervals as a motivation, the distinction between *continuous-time Markov chains (CTMC)*, *continuous control with continuous action spaces*, and *discrete-time approximations* is **not clearly articulated**.
It remains unclear **why a CTMC-based formulation is fundamentally necessary**, and what benefits it provides over well-established discrete-time methods with sufficiently small time steps. The experimental tasks are sufficient using the discrete-time methods enough.
This conceptual gap makes the overall significance of the work questionable.

### (2) Insufficient discussion on relation to model-based MARL

The proposed VIP framework involves explicit learning of **a dynamics model** ( f_\psi(x, u) ) and a **reward model** ( r_\phi(x, u) ), followed by value and policy optimization — characteristics more aligned with **model-based RL** rather than model-free methods.
However, the paper **does not compare or discuss** how VIP relates to representative **model-based multi-agent RL** or **model predictive control (MPC)** approaches (e.g., MA-MBRL, MA-MPC, or recent differential game solvers).
This omission weakens the positioning of the work.

### (3) Limited baseline comparisons

Experimental baselines are limited to other continuous-time algorithms (e.g., HJBPPO, DPI, IPI).
However, for a fair evaluation in MARL, it is crucial to include **widely recognized baselines** such as **SMPE**[1], **Revisiting Off-policy MARL**[2], and **RACE**[3], which remain standard for multi-agent benchmarks like MPE and MuJoCo.
Without such comparisons, it is difficult to assess whether improvements stem from the continuous-time formulation or from differences in model complexity.

[1]SMPE: Enhancing Cooperative Multi-Agent Reinforcement Learning with State Modelling and Adversarial Exploration. ICML 2025

[2]Revisiting Cooperative Off-Policy Multi-Agent Reinforcement Learning. ICML 2025

[3]RACE: Improve Multi-Agent Reinforcement Learning with Representation Asymmetry and Collaborative Evolution. ICML 2023



### (4) Lack of intuitive explanation and algorithmic clarity

The paper is mathematically dense but lacks intuitive explanation.
For instance, the **motivation and role of VGI** are underexplained in the main text, and there is little visualization or step-by-step discussion of the **training procedure**, **computational cost**, or **failure cases**.
Similarly, the algorithm section lacks a high-level intuition connecting the PDE-based reasoning with practical RL optimization.
Readers unfamiliar with HJB theory or PINNs may find it hard to grasp the conceptual novelty.

### (5) Writing and presentation quality

Although mathematically complete, the paper suffers from **dense notation, unclear transitions, and insufficient narrative coherence**.
Sections often read like technical reports rather than structured scientific arguments.
The connection between the theoretical motivation and empirical evidence needs stronger integration.

While the paper presents an interesting integration of **PINNs** and **continuous-time MARL**, it lacks clear motivation, comprehensive baseline coverage, and intuitive exposition.
The conceptual distinction between continuous-time and discrete-time MARL remains ambiguous, and the empirical validation does not convincingly demonstrate broad significance.
Substantial revisions — including stronger justification, clearer writing, and broader experimental evaluation — are necessary before this work could be considered for publication at ICLR.

**Questions:**

To make the paper stronger and clearer, the following points should be addressed:

1. **Motivation of continuous-time formulation:**
   Why is the continuous-time formulation superior to a fine-grained discrete approximation?
   Can you clarify the distinction between CTMC and continuous-action control tasks?

2. **Connection to model-based MARL:**
   Since VIP involves explicit modeling of ( f_\psi ) and ( r_\phi ), how does it differ from model-based methods such as MA-MBRL or MA-MPC?
   A comparative discussion (or experiment) is needed.

3. **Experimental coverage:**
   Can you include or at least discuss results against mainstream MARL baselines (e.g., SMPE, Revisiting Off-policy MARL, MAPPO)?
   This would better contextualize the practical relevance of your approach.

4. **Algorithmic intuition:**
   Could you provide an intuitive explanation or figure illustrating how the VGI module refines the gradient flow and why it is necessary for training stability?

5. **Scalability and computation:**
   What is the computational overhead of solving the PINN-based HJB equations in multi-agent systems?
   Can this method scale to more than 10 agents or to real-time control?

6. **Clarify novelty:**
   Many parts of the method (PINN + HJB + actor-critic) exist in prior continuous-time single-agent works.
   Please clearly highlight what is *new* in your multi-agent extension, both conceptually and technically.

---

> ### Author Response · Authors · 2025-11-24
> **Thanks for the constructive feedback, we provide detailed responses below: Reply 1**
>
> We appreciate the reviewer's time and detailed comments. We carefully address each concern below with clarifications.
>
> **W1. Motivation and conceptual clarity**
>
> (1) Our paper does not rely on CTMC, but follows the standard continuous-time optimal control setting. A CTMC describes a stochastic jump process with a generator matrix. Time evolves continuously, but the state evolves through discrete jumps governed by exponential waiting times. CTMCs are primarily used for discrete-state stochastic jump processes like queuing systems. To the best of our knowledge, there is no CTRL literature that adopts CTMC as its theoretical basis. Our formulation follows the classical continuous-time control model used in optimal control and HJB theory, which is also well-adopted in CTRL. We will further clarify the differences to avoid any confusion in the future version.
>
> (2) We agree that further strengthening the motivation can make it easier for the readers. We have revised the first paragraph of the Introduction (marked blue), and we have also clarified here: DTRL approximates a continuous-time process by imposing a fixed discretization step. This introduces two inherent limitations. (i) If the timestep is coarse, the resulting controller becomes non-smooth and leads to suboptimal or unstable behavior. (ii) If the timestep is fine, the number of states and iteration steps becomes large, which enlarges not only memory storage but also many learning trials. Besides, the Bellman operator will degrade as the time step $\Delta t$ approaches $0$, since the target of temporal difference can be dominated by approximation noise, causing the Bellman updates to become unstable with high variance.
>
> **W2. Insufficient discussion on relation to model-based MARL**
>
> Indeed, we use model-based RL to conduct our algorithms. We have highlighted in Section 4.1 that DPI, ODE are model-based baselines, while IPI remains model-free. These model-based methods explicitly learn (or require) system dynamics and therefore fall exactly into the same category as our algorithm. Please refer the Section 4.1 for more details.
>
> **W3. Limited baseline comparisons**
>
> We thank the reviewer for pointing out these additional references. After carefully reading [1], [2] and [3]. We found that these methods rely on specialized architectural or structural assumptions (e.g., state modeling modules, adversarial exploration) which are out of our scope.
> Instead, to more clearly assess whether our performance gains come from the continuous-time formulation or model complexity, we compare against widely recognized standard discrete-time MARL baselines—MADDPG, MATD3, and MAPPO.
> Specifically, Figure 9 shows that our method outperforms these discrete-time baselines in continuous-time environments, whereas Figure 13 shows that this advantage diminishes when the task becomes effectively discrete-time. These results clarify the source of our performance improvements. (1) Figures 2 and 4 demonstrate that, compared to other CTRL methods, our gains stem from architectural advantages: the PINN+VGI yields more accurate value and value-gradient approximations, which explicitly improve policy learning. (2) Figure 9 and 13 show that, compared to discrete-time algorithms, our method benefits from being designed specifically for continuous-time dynamics, explaining why the gain is pronounced in continuous-time settings but vanishes when the environment behaves like a discrete-time system.
>
> **W4. Lack of intuitive explanation and algorithmic clarity**
>
> To improve clarity, several revisions have been incorporated in our revised paper. In the introduction, we added a more explicit explanation of why discrete-time approximations are not efficient. At the beginning of the Methodology section, we included a high-level summary to guide readers before entering the technical details. In addition, intuitive explanations have been added before Lemma 3.3 and Theorem 3.4, clarifying their purpose and practical significance for policy improvement and value gradient stability.
>
> For the mathematical content, Appendix A now begins with a structured guidance section describing the goal of each lemma and theorem, so that readers can follow the theoretical components without getting lost in derivations. The training algorithm is provided in Algorithm 1 (Appendix A.8), and the computational cost (time, CPU/GPU memory usage) is reported in Appendix E.
>
> Regarding the intuition of VGI: PINN-based value learning enforces HJB residuals but does not guarantee accurate value gradients. Since policy learning depends on the value gradients, the errors can explicitly degrade the performance. The VGI module addresses this by propagating value-gradient information along sampled trajectories and enforcing consistency with the dynamics Jacobian. This creates a self-correcting target for $\nabla_x V$ that complements the PDE constraint, improves the accuracy of value gradients, and thus improves performance.

---

> ### Author Response · Authors · 2025-11-24
> **Reply 2**
>
> **Q1. Motivation of continuous-time formulation**
>
> We have provided a detailed explanation in our response to **W1**. Please refer to that response for the complete discussion.
>
> **Q2. Connection to model-based MARL**
>
> We have provided a detailed explanation in our response to **W2**. Please refer to that response for the complete discussion.
>
> **Q3. Experimental coverage**
>
> We have provided a detailed explanation in our response to **W3**. Please refer to that response for the complete discussion.
>
> **Q4. Algorithmic intuition**
>
> The intuition behind VGI is listed as follow: PINN-based HJB residuals constrain the value function $V_\theta$ at the function level but do not guarantee accurate value gradients. Since PDE and policy learning depend on the value gradients, the errors can be amplified in subsequent learning, thus degrading the performance. The VGI module corrects this by propagating gradient information along observed trajectories. Specifically, the update
> $g_{t} = \nabla_x r(x_t,u_t) \Delta t + e^{-\rho\Delta t} \nabla_x f(x_t,u_t)^\top \nabla_x V_\theta(x_{t+\Delta t})$
> enforces local consistency between the value gradient, the reward landscape, and the system dynamics. This forms a trajectory-aligned gradient flow that explicitly regularizes $\nabla_x V_\theta$ in directions that the PINN residual alone cannot control.
>
> We demonstrate the effectiveness of VGI in Figure 3 and 10, which show the difference with and without VGI. The values and value gradients are much more accurate with the VGI module, which validates our claim.
>
> **Q5. Scalability and computation**
>
> The computation cost is listed in Table 3 (Appendix A.8). In summary, VIP has comparable training time and resource usage to other model-based continuous-time methods, including VGI, DPI, ODE, and HJBPPO. IPI, MATD3, and MAPPO exhibit slightly shorter runtimes because they do not require solving continuous-time dynamics or HJB-style losses. MADDPG is the fastest, as it is a lightweight model-free method without any additional modules needed. For the scalability, our method has been evaluated on tasks with up to 113-dimensional states and 6 agents.
>
> **Q6. Clarify novelty**
>
> The key challenge in extending continuous-time PINN--HJB methods to multi-agent settings is not the HJB formulation itself, but the instability of value-gradient learning. In single-agent CTRL, PINN residuals are usually sufficient to obtain reasonably stable value approximations. However, in multi-agent systems, the dynamics Jacobian $\nabla_x f$ couples agents' states. Even small errors in $\nabla_x V$ are amplified across agents, which leads to degraded policy training (Please refer the Figure7 and Figure 10).
>
> Experimentally, we also verify the multi-agent nature of this contribution: in Figure 12, we conduct Cooperative MARL vs Single-Agent RL, comparing three algorithmic perspectives on the coupled oscillator environment as the number of agents increases from $N=2$ to $N=6$ ($N=2,4,6$):  (1) *VIP* (ours),  (2) *VIP-SA*, which treats the entire multi-agent system as a single  high-dimensional agent, and  (3) *VIP-DEC*, a fully decentralized variant without a centralized critic, while replaced by decentralized critics for each agent. When $N=2$, all three variants obtain reasonably strong performance because the dimensionality of the joint action space is low and the dynamics remain fairly simple. But as the number of agents increases, the differences between these learning paradigms become pronounced. VIP-SA forces the learner to operate in a joint action space of dimension $N$, which grows exponentially in exploration complexity. This makes policy updates inefficient and leads to significantly degraded performance. In contrast, VIP-DEC avoids this scaling issue but suffers from severe non-stationarity. VIP maintains high performance for $N=4$ and $N=6$, since VIP avoids both issues by centralizing value learning while keeping policies decentralized, the critic learns a consistent continuous-time value function that captures the full joint dynamics without exposing individual policies to exploding action spaces.

---

### Official Review · Reviewer_jF5y · 2025-10-31

**Soundness:** 4
**Presentation:** 4
**Contribution:** 4
**Rating:** 10
**Confidence:** 3

**Summary:**

This paper proposes a Continuous-Time Mutli-Agent RL algorithm based on Physics-Informed Neural Networks (PINN). The algorithm, called Value Iteration via PINN (VIP), employs Value Gradient Iteration to estimate the gradient of the value function represented as HJB PDEs. A model of the environment and a reward model are trained alongside policy and value function, and are used for estimating the gradients for value function and expected advantage with respect to the critic and policy parameters respectively. Value gradient iteration, a method for refining value function gradient estimates, is introduced and used for computing an additional target for the value function. The algorithm is evaluated empirically and shown to be competitive with a diverse set of recently introduced methods. Finally, an ablation assesses the effect of individual design choices.

**Strengths:**

The paper has a great buildup with excellent motivation including a succinct discussion of related work. Problem formulation and other preliminaries are presented cleary. The central theoretical claims all come with a proof. The value gradient iteration (VGI) method introduced in this paper may be of broad interest for CT-RL in general. Utilizing the differential nature of PINNs in order to train them on more refined targets that include the estimated gradient is a great application of this kind of model. A diverse set of environments is used for evaluation and the experimental results look promising. The effectiveness of VGI is shown empirically on a toy example.

Overall, the paper was very pleasant to read and the contribution to the ICLR community is likely very signficant.

**Weaknesses:**

No discrete-time RL baseline was included in the experiments. Y-axis scaling in some of the plots make them difficult to read.

**Questions:**

- The results on the walker environment show a significantly higher cumulative reward for VIP when compared to the baselines. Moreover, the cumulative reward drops significantly with further training. Can you explain this?
- For the cooperative navigation task many of the compared algorithms show a very large variance which implies that for some seeds the performance is much better. Can you elaborate on that?

---

> ### Author Response · Authors · 2025-11-24
> **Thank you for your valuable comments. Our detailed responses are provided below.**
>
> Many thanks to your careful evaluation and highly constructive comments. The questions raised have helped us further strengthen the clarity and completeness of the work. We provide detailed, point-by-point responses below.
>
> **Q1. No discrete-time RL baseline was included in the experiments.**
>
> We added MATD3, MADDPG, and MAPPPO as discrete-time baselines. Because Figures 2 and 4 already contain six curves, we report these additional baselines separately in Figure 9 to keep the plots readable. All discrete-time methods' performance degrades in these tasks. But this degradation is expected: discrete-time Bellman backups implicitly assume fixed time steps, and their value approximations become inaccurate when the underlying system evolves in continuous time with arbitrary time intervals.
>
> For completeness, we also evaluate our algorithms in discrete-time settings in Figure 13 (Appendix C.4). The experimental results still show that our algorithms perform comparably with these discrete-time baselines. Crucially, the previous performance gains observed in our main experiments diminish under fixed-step discrete-time dynamics, indicating that our advantages stem from modeling continuous-time formulation directly.
>
> **Q2. Y-axis scaling in some of the plots makes them difficult to read.**
>
> Thank you for pointing this out. We have updated the scaling of the Cooperative Navigation plots (Figure 4 and Figure 7) to avoid over-compression of the learning curves.
>
> **Q3. Explanation about Performance in Walker**
>
> (1) In the walker environment, all compared baselines operate in continuous time, but none of them have modules for correcting the value-gradient errors to help improve the output accuracy of the critic. In our methods, VIP leverages the VGI module to dynamically correct the value gradients, which further improves the performance. Specifically, the ablated variant of VIP (VIP w/o VGI) also performs weaker in the Walker environment, which indicates the effectiveness of VGI.
>
> (2) In the finite-time settings, PINN is usually set as time-dependent, which can leverage terminal/boundary loss to further stabilize the training process. However, MuJoCo environments do not naturally provide meaningful terminal or boundary conditions, and these conditions are inherently unavailable in our infinite-horizon formulation, making terminal loss–based stabilization inapplicable. Therefore, it can lead to some degraded performance during subsequent training.
>
> **Q4. Explanation about Performance in Cooperative Navigation.**
>
> Thank you for the insightful observation. Our environments in this setting are deterministic. In such cases, if the critic’s value or value-gradient approximations are inaccurate early in training, the policy may quickly converge to a suboptimal equilibrium and remain trapped there, since no substantial stochasticity exists to help the agent escape. This phenomenon is particularly pronounced in Cooperative Navigation since in these settings, the target for each agent is not fixed; each agent implicitly selects the nearest landmark as its target. As a result, every seed may converge to different results. In contrast, our method uses the VGI module to continuously correct value-gradient approximations, leading to more consistent coordination in this task.

---

> > ### Comment · Reviewer_jF5y · 2025-11-26
> >
> > I have read the rebuttal and thank the authors for their clarifications. I keep my assessment unchanged.

---

> > > ### Author Response · Authors · 2025-11-29
> > > **Thanks**
> > >
> > > We sincerely thank the reviewer for the time and effort devoted to evaluating our work, and we truly appreciate your thoughtful assessment.

---

### Official Review · Reviewer_5Xhd · 2025-11-02

**Soundness:** 2
**Presentation:** 2
**Contribution:** 2
**Rating:** 2
**Confidence:** 4

**Summary:**

This paper extends prior work in continuous-time reinforcement learning (CTRL) from single-agent settings to cooperative multi-agent reinforcement learning (MARL). The authors focus on overcoming the challenges of large state dimensionality and accurate value approximation in continuous domains.
Their proposed framework combines Physics-Informed Neural Networks (PINNs) with a new Value Gradient Iteration (VGI) module.

- PINNs are used to approximate the viscosity solution of the Hamilton–Jacobi–Bellman (HJB) equation in continuous time, addressing the curse of dimensionality without explicit discretization.

- The VGI module refines value gradients through a recursive, single-step Bellman-like update, which improves stability and accuracy in policy learning.


The paper presents both theoretical analysis (via a contraction-based convergence argument) and empirical validation on continuous-time adaptations of the Multi-Agent Particle Environment (MPE) and Multi-Agent MuJoCo benchmarks.

**Strengths:**

- Scalability to high-dimensional systems: The integration of PINNs with VGI addresses the curse of dimensionality in continuous-time settings, enabling effective learning in systems with large state spaces. To my knowledge, VGI module is a novel contribution
- Benchmarking contribution: The authors extend two widely-used discrete MARL benchmarks—Multi-Agent Particle Environment (MPE) and Multi-Agent MuJoCo—to continuous-time settings.

**Weaknesses:**

I have previously reviewed this paper for another conference, and while the authors addressed some feedback in this version, several substantive issues remain unaddressed, which I list here:

- **Scope is limited to Cooperative MARL & lacks comparison with relevant baselines.** Although framed as a CT-MARL approach, the method applies only to cooperative continuous-time MARL scenarios. It does not handle competitive or mixed settings, where agents have conflicting objectives, which would require solving Hamilton–Jacobi–Isaacs (HJI) equations rather than HJB equations. This limits the generality of the contribution—especially because many real-world multi-agent systems involve competition or partial cooperation. Moreover, in the cooperative case, the learning problem effectively reduces to solving a single centralized control problem with a large joint state space. This raises the question: Is the proposed method genuinely multi-agent in nature, or simply a high-dimensional continuous control solver using a centralized critic? Thus, in my understanding, the high dimensionality of the state could also be an issue in CT-RL. This raises the need for a more thorough comparison with single-agent CT-RL methods.

- **Incremental Theoretical Contribution.** The convergence result (Theorem 3.4) relies on a standard contraction mapping argument under bounded Jacobians and time-invariant dynamics. While mathematically sound, it does not introduce new theoretical insight specific to MARL or continuous-time value iteration. The proof does not incorporate any properties unique to cooperative or decentralized MARL (such as non-stationarity, joint-policy coupling, or inter-agent coordination). Thus, it serves more as a consistency check, rather than a result illuminating how the number of agents or their interactions affect stability or learning dynamics.

- **Motivation and Framing**. The motivation is overall clear—discrete-time MARL algorithms degrade under variable time steps—but the experimental evidence for this claim remains limited. Figure 1 provides a useful visual demonstration, but it’s unclear whether the performance drop of discrete methods (e.g., MADDPG) reflects fundamental limitations of discrete-time updates or simply under-tuned baselines; see next comment. This concern was raised by multiple NeurIPS reviewers and remains only partially addressed.

- **Methodological Comment**. It would strengthen the empirical evaluation if the proposed method were also tested on standard discrete-time benchmarks where the baseline algorithms (e.g., MAPPO, MADDPG) are known to perform well and reproducibly, ideally reproducing reported results from the original papers. This would help disentangle whether the observed advantages stem from the continuous-time formulation itself or from known optimization and reproducibility issues in discrete MARL. Since the proposed approach is theoretically a continuous-time generalization of Bellman updates, it should in principle also perform competitively in discrete settings when the time step is small.

- **Limited Experiments.** In summary, the empirical comparison still omits several key baselines and comparative analyses: no competitive or stochastic control baselines, no detailed compute or resource comparison (training time, memory usage), discrete-time baselines (MADDPG, MAPPO, MATD3) are still missing or not well-tuned for variable time steps.

Overall, the authors should better articulate what is new in the multi-agent context and/or appropriately compare the proposed method.


## Minor Comments


- In 3.4.2, they still mention the use of terminal-condition losses, while they don’t explicitly exist in equation (17) anymore, which makes sense if the paper now addresses the infinite horizon setup that makes it not ideal to compute such a loss. This should be clarified—either by explicitly removing the term from the text or explaining how those losses were absorbed into the loss formulation.
- The introduction should clearly state that the method applies only to deterministic multi-agent systems.
- a brief note distinguishing the smooth vs. viscosity case would help prevent confusion.

**Questions:**

1. Can you provide empirical or analytical evidence that the method provides benefits specifically because of its multi-agent nature (e.g., coordination effects), rather than just being a strong centralized continuous control solver?
2. Computational Cost: Could you report the relative compute cost (training time, memory, convergence rate) of VIP compared to DPI, IPI, or ODE baselines?
3. Stochastic Dynamics: Would your framework still hold if small stochastic noise were introduced in the dynamics (i.e., moving toward stochastic differential equations)?
If not, how sensitive is the PINN–VGI coupling to such noise?
4. Value Gradient Iteration Stability: The VGI recursion resembles a single-step gradient rollout. Does it ever introduce instability or oscillation during training? Are there hyperparameters controlling its contraction rate?

---

> ### Author Response · Authors · 2025-11-24
> **Thanks for taking the time to review our work: reply 1**
>
> Firstly, we sincerely thank the reviewer for taking the time to review our paper again. We thank the reviewer for giving more detailed and constructive comments. We have revised the paper corresponding to your suggestions, and the response to your comments are listed below:
>
> **W1. Scope is limited within Cooperative MARL Settings.**
>
> We acknowledge that our paper focuses on cooperative settings, and we have clearly shown that in the Abstract, Introduction, and Methodology sections. Extending the framework to competitive settings would require a fundamental change in the problem formulation, since it would involve solving HJI equations instead of HJB equations. This reformulation is beyond the scope of the current paper. We highlighted in the Conclusion that extending our framework to HJI-based competitive settings is an important next step.
>
> **W2. Is the proposed method genuinely multi-agent in nature, or simply a high-dimensional continuous control solver using a centralized critic?**
>
> In the revised manuscript, we design a new experiment to demonstrate that our algorithms are multi-agent in nature. In Figure 12 of Appendix C.3, we conduct Cooperative MARL vs Single-Agent RL, comparing three algorithmic perspectives on the coupled oscillator environment as the number of agents increases from $N=2$ to $N=6$ ($N=2,4,6$): (1) *VIP* (ours), (2) *VIP-SA*, which treats the entire multi-agent system as a single high-dimensional agent, and (3) *VIP-DEC*, a fully decentralized variant without a centralized critic, while replace by decentralized critics for each agent. When $N=2$, all three variants obtain reasonably strong performance because the dimensionality of the joint action space is low and the dynamics remain fairly simple. But as the number of agents increases, the differences between these learning paradigms become pronounced. VIP-SA forces the learner to operate in a joint action space of dimension $N$, which grows exponentially in exploration complexity. This makes policy updates inefficient and leads to significantly degraded performance. In contrast, VIP-DEC avoids this scaling issue but suffers from severe non-stationarity. VIP maintains high performance for $N=4$ and $N=6$, which confirms its multi-agent nature. This is because VIP avoids both issues by centralizing value learning, thereby mitigating the non-stationarity, and it uses decentralized continuous-time policies rather than treating the system as a high-dimensional single agent.
>
> **W3. Incremental Theoretical Contribution: lacks MARL insight.**
>
> We would like to clarify that our proof follows the standard theoretical treatment of MARL. In the cooperative setting, the environment is modeled as a joint dynamical system, and convergence is typically established by defining a Bellman(-type) operator on this joint space and proving that it is a contraction. This proof pattern is widely used in MARL theory (e.g., [1], [2], [3]). To avoid any ambiguity, we explicitly highlight in the proof of Theorem 3.4 that the dynamics are joint multi-agent dynamics, consistent with the standard treatment in cooperative MARL.
>
> [1] Li, S., Guo, J., Xiu, J., Xu, R., Yu, X., Wang, J., ... & Liu, X. Byzantine robust cooperative multi-agent reinforcement learning as a bayesian game. ICLR 2024.
>
> [2] Yang, Y., Luo, R., Li, M., Zhou, M., Zhang, W., & Wang, J. Mean field multi-agent reinforcement learning. ICML 2018.
>
> [3] Wang, J., Ren, Z., Han, B., Ye, J., & Zhang, C. Towards understanding cooperative multi-agent q-learning with value factorization. NIPS 2021.
>
> **W4. Limited Experiments with the lack of the discrete-time baselines**
>
> We agree that adding more discrete-time baselines can help readers understand better. Based on the reviewers' suggestions, we added three baselines: MADDPG, MAPPO, and MATD3. Our results have been displayed in Figure 9, which is marked blue in the experimental part. All discrete-time methods' performance degrades in these tasks. But this degradation is expected: discrete-time Bellman backups implicitly assume fixed time steps, and their value approximations become biased or unstable when the underlying system evolves in continuous time with arbitrary or environment-dependent time intervals.
>
> Additionally, we test our algorithms in discrete-time settings in Figure 13 of Appendix C.4. The results demonstrate that our method remains comparable to these discrete-time baselines in their native regime but loses the significant gains while in the continuous-time settings, confirming that our advantages are specific to continuous-time environments.

---

> ### Author Response · Authors · 2025-11-24
> **Reply 2**
>
> **W5.  Minor Comments: terminal loss in section 3.4.2**
>
> Thanks for pointing out the typo. Since this is not a finite-time reinforcement learning, we deleted the terminal loss for the critic learning.
>
> **W6. The introduction should clearly state that the method applies only to deterministic multi-agent systems.**
>
> We add this part in the contribution of the Introduction section to prevent any confusion. Please check the blue-marked part in the revised manuscript.
>
> **W7. A brief note distinguishing the smooth vs. viscosity case would help prevent confusion.**
>
> We have added this note in Appendix D.1 of the revised paper. Specifically, the HJB equation admits both smooth and viscosity formulations. Smooth solutions require the value function to be differentiable and satisfy the PDE equations, while viscosity solutions handle non-smooth value functions using sub/super-differential conditions. In our paper, we focus on the smooth solutions; extending VIP to non-smooth or stochastic settings would require viscosity-solution theory and is left for our future work.
>
> **Q1. Empirical or analytical evidence that the method provides benefits specifically because of its multi-agent nature**
>
> We have replied a similar question in the previous section **W2**. Please refer to the response to **W2** for the full explanation.
>
> **Q2. Computational Cost**
>
> We provide the computational cost in Table 3 in Appendix E of the revised paper. Specifically, the table reports the training time (hours), CPU memory usage (GiB), and GPU memory usage (MiB) for all baselines (VIP, DPI, IPI, ODE, HJBPPO, MADDPG, MATD3, MAPPO) across all evaluated environments.
>
> In summary, VIP has comparable training time and resource usage to other model-based continuous-time methods, including VGI, DPI, ODE, and HJBPPO. IPI, MATD3, and MAPPO exhibit slightly shorter runtimes because they do not require solving continuous-time dynamics or HJB-style losses. MADDPG is the fastest, as it is a lightweight model-free method without any additional modules needed.
>
> **Q3. Stochastic Dynamics: Would your framework still hold if small stochastic noise were introduced in the dynamics?**
>
> Although our framework focuses on deterministic settings, the learned model does not collapse under small noise/disturbance. Specifically, we inject different levels of noise into the dynamics model, i.e. $x_{t+\Delta t} = f(x_t, u_t)\Delta t + \varepsilon_t, \varepsilon_t \sim \mathcal{N}(0,\sigma^2 I)$. We have tested $\sigma^2=0.1$ (Low), $0.5$ (Mid), and $1.0$ (High) settings. And the experimental results in Figure 14 (Appendix C.5) show that the performance is still robust when the noise level is low. However, the performance degrades sharply in the mid and high noise settings. This behavior is expected. Both the HJB-based critic and the VGI update rely on assumptions that hold only under deterministic or low-variance dynamics (e.g., well-defined state derivatives and smooth value gradients). When stochasticity becomes large, these assumptions are violated, leading to biased PDE residuals and unreliable gradient propagation.
>
> **Q4. Value Gradient Iteration Stability**
>
> VGI is designed to stabilize convergence by enforcing gradient-level consistency with the continuous-time dynamics. Actually, based on the experimental results, the VIP consistently outperforms VIP w/o VGI, and we observe that VGI helps to converge but does not introduce instability in practice. The contraction rate is primarily controlled by (i) the discount factor $\rho$, (ii) the integration step $\Delta t$, and (iii) the critic learning rate.
>
> We sincerely thank the reviewer for taking the time to re-evaluate our submission and for providing thoughtful feedback. We truly appreciate the opportunity for further back-and-forth discussion, and we hope our clarifications help address the reviewer’s concerns. Thank you again for your time and consideration.

---

### Meta-Review · Area_Chair_twNk · 2026-01-08

**Summary:**

This paper proposes a continuous-time multi-agent reinforcement learning framework using Physics-Informed Neural Networks to solve Hamilton-Jacobi-Bellman equations. While the technical approach is interesting, the reviewers identified several fundamental and unresolved weaknesses that significantly undermine the paper's contribution and readiness for publication.

The primary concerns are:

Incremental Theoretical Contribution: The convergence proof relies on standard arguments and does not provide new insight specific to MARL or continuous-time value iteration. It fails to illuminate how multi-agent interactions affect learning.

Unconvincing Motivation and Empirical Validation: The core motivation—that discrete-time MARL degrades under variable time steps—is not sufficiently supported. The experimental comparison was initially lacking standard discrete-time MARL baselines (e.g., MADDPG, MAPPO, MATD3) and key multi-agent baselines from recent literature. The empirical design does not convincingly isolate the benefit of the continuous-time formulation from other factors.

**Reviewer Concerns:**

Addressed Concerns:

Discrete-Time Baselines: The authors have added comparisons with MATD3, MAPPO, and MADDPG (new Figs. 9, 13). This is a necessary but insufficient improvement.

Robustness to Stochasticity: Experiments with injected stochastic noise (new Fig. 14) partially address a methodological question.

Clarification on CTDE: Additional discussion and an experiment (new Fig. 12) aim to emphasize the multi-agent, CTDE nature of the framework.

Outstanding Concerns:

Scope and Generality: The framework's inapplicability to competitive or mixed settings (requiring HJI equations) and its conceptual similarity to a centralized solver remain severe limitations that constrain the paper's significance for the broader MARL community.

Theoretical Novelty: The authors did not substantively address the criticism that the theoretical contribution is incremental.

**Reviewer Scores:**

Reviewer 5Xhd (Initial: 2 - Reject): Estimated Final Score: 2 (Reject).

Reviewer qN3L (Initial: 2 - Reject): Estimated Final Score: 2 (Reject).

Reviewer b9mm (Initial: 6 - Borderline Accept) Estimated Final Score: 4 or 6 (Borderline Reject or Borderline Accept).

Reviewer jF5y (Initial: 10 - Strong Accept): Estimated Final Score: 10 (Strong Accept).

---

### Decision · Program_Chairs · 2026-01-26

Accept (Poster)